# Photoresponsive spiro-polymers generated in situ by C–H-activated polyspiroannulation

Ting Han [1,2,3], Zhanshi Yao [4], Zijie Qiu[1,2], Zheng Zhao[1,2], Kaiyi Wu[4], Jianguo Wang[1,2], Andrew W. Poon [4], Jacky W.Y. Lam[1,2]* & Ben Zhong Tang [1,2,3]*

The development of facile and efficient polymerizations toward functional polymers with unique structures and attractive properties is of great academic and industrial significance. Here we develop a straightforward C–H-activated polyspiroannulation route to in situ generate photoresponsive spiro-polymers with complex structures. The palladium(II)-catalyzed stepwise polyspiroannulations of free naphthols and internal diynes proceed efficiently in dimethylsulfoxide at 120 °C without the constraint of apparent stoichiometric balance in monomers. A series of functional polymers with multisubstituted spiro-segments and absolute molecular weights of up to 39,000 are produced in high yields (up to 99%). The obtained spiro-polymers can be readily fabricated into different well-resolved fluorescent photopatterns with both turn-off and turn-on modes based on their photoinduced fluorescence change. Taking advantage of their photoresponsive refractive index, we successfully apply the polymer thin films in integrated silicon photonics techniques and achieve the permanent modification of resonance wavelengths of microring resonators by UV irradiation.

[1] HKUST Shenzhen Research Institute, No. 9 Yuexing 1st RD, South Area Hi-tech Park, Nanshan, Shenzhen 518057, China. [2] Department of Chemistry, Hong Kong Branch of Chinese National Engineering Research Center for Tissue Restoration and Reconstruction, Institute for Advanced Study, and Department of Chemical and Biological Engineering, The Hong Kong University of Science and Technology, Clear Water Bay, Kowloon, Hong Kong, China. [3] Center for AIE Research, College of Materials Science and Engineering, Shenzhen University, Shenzhen 518060, China. [4] Department of Electronic and Computer Engineering, The Hong Kong University of Science and Technology, Clear Water Bay, Kowloon, Hong Kong, China. *email: chjacky@ust.hk; tangbenz@ust.hk

The development of facile and efficient synthetic strategies toward functional polymers with unique structures, attractive properties and advanced applications is of both academic interest and practical implication. Among various functional polymers, photoresponsive polymers that can alter their optical properties in response to light irradiation have drawn great attention over recent decades due to the increasing demands on smart optical and biological materials[1–3]. Based on their different light-sensitive optical properties, photoresponsive polymers can be used in diverse advanced applications[4]. For example, polymers with light-responsive absorption or fluorescence properties have been used in smart surfaces, optical sensors, optical or photoelectric switches, and information storage[5–8]. Polymers with light-regulated refractive index are useful in fine-tuning the properties of phase-sensitive optical devices[9]. Different from other commonly used stimuli such as temperature, pH, and electric and magnetic field, light irradiation allows easy and precise control on the temporal and positional resolution of the response in a non-contact and remote way. The strength of the response can be finely modulated by controlling the light irradiation parameters such as light intensity and irradiation time. Meanwhile, compared with small molecule analogs, the good processability and film-forming ability of polymers can facilitate the manufacture process and enable materials to achieve their photoresponsiveness in engineering robust conditions. Therefore, it is highly desirable to develop and synthesize photoresponsive polymer materials and explore their potential applications.

The construction of photoresponsive polymers are generally achieved either by physically blending well-known photoresponsive molecules in polymer matrices or by chemically incorporating photoresponsive units into polymer side chains or main chains via different polymerization techniques[4]. Among them, step-growth polymerizations of $A_2 + B_2$ monomers (where A and B are mutually-reactive groups) play an indispensable role in the synthesis of main-chain photoresponsive polymers. There are two fundamental requirements for these polymerizations to produce high molecular weight polymers with good performance. One is the inevitable use of two difunctional monomers, and the other is a strict control on the stoichiometric balance of monomers[10–12]. However, difunctional monomers with photoresponsive moieties such as difunctionalized spiropyrans and coumarins often suffer from limited structural variety and synthetic difficulty[13–15]. Furthermore, the monomer stoichiometric conditions are hard to be met in practice due to the presence of side reactions, impurities in monomers and solvents, evaporation or decomposition of monomers, precipitation of polymer segments, etc. Then one question arises: is it possible to develop a stepwise polymerization strategy that can in situ generate photoresponsive polymers using commercially available or naturally occurring monofunctional monomers and meanwhile without the constraint of apparent stoichiometric balance in monomers?

As C–H bonds are ubiquitous in various organic compounds, it would be desirable if the C–H bond can serve as a hidden functional group in one monomer to participate in the polymerization process. Inspired by the rapid progress of C–H activation in organic chemistry and conjugated polymer synthesis[16–22], we envision that stepwise polymerizations based on the transition metal-catalyzed C–H activation will be a good choice to transform the abundantly existing but poorly reactive monofunctional monomers into valuable photoresponsive polymers. Meanwhile, the employment of C–H as the reactive functional group may allow the corresponding polymerizations to be more tolerant to the mismatch in functional group stoichiometry than conventional step-growth polymerizations[23–27]. Recently, an interesting palladium(II)-catalyzed spiroannulation reaction of free naphthols and alkynes have captured our attention. As depicted in Fig. 1a, this reaction can facilely transform the commercially available and inexpensive 2-naphthols into complex spirocyclic structures based on a joint C–H functionalization/dearomatization strategy[28]. Since double bonds play an important role in many photoresponsive materials such as stilbenes, butadienes, anthracenes, (thiophene) fulgides, coumarins, and cinnamic esters[5,29], the spirocyclic structure with multiple C=C bonds especially with the carbonyl-activated C=C is thus anticipated to be a potential photoresponsive structure unit. In addition, the rigid spirocyclic structure may also endow the corresponding polymers with unique properties such as high thermal and morphological stability[30].

With these considerations in mind, herein we explore the possibility of developing this C–H-activated spiroannulation reaction into a straightforward polymerization strategy to in situ generate photoresponsive spiro-polymers from simple monofunctional monomers (Fig. 1b). The C–H-activated polyspiroannulations of commercially available naphthols and readily available diynes proceed efficiently in the presence of excess naphthols. A series of multisubstituted spiro-polymers with high molecular weights (absolute $M_w$ up to 39,000) are generated in situ in high yields (up to 99%). Thanks to its unique reaction mechanism, this step-growth polymerization can be used for the preparation and further polymerizations of telechelic polymers. The obtained spiro-polymers show excellent film-forming ability, high thermal and morphological stability, and interesting photoresponsive optical properties. The light-induced fluorescence conversion between the bright and dark states of these polymers enables the generation of well-resolved fluorescent photopatterns in different working modes. Furthermore, the UV-tunable refractive index of the polymer thin films is successfully applied for the permanent modification of resonance wavelengths of microring resonators.

## Results

**Polyspiroannulation**. To develop the C–H-activated polyspiroannulations, commercially available 2-naphthol (**1a**) and quinolin-6-ol (**1b**) and readily accessible internal diynes **2a–e** were selected as monomers (Fig. 1b and Supplementary Fig. 1)[25,31]. The polymerization conditions were first systematically optimized by investigating the effect of catalyst type and loading, solvent, reaction temperature, monomer feed ratio, additive and oxidant loading, monomer temperature, and time course on the polymerizations of **1a** and **2a** (Table 1 and Supplementary Tables 1–5). The time course of the polymerization demonstrates the high efficiency of this polymerization (Supplementary Table 5). A polymer with a relative number-average molecular weight ($M_n$) of 9500 was produced in a good yield of 76% after reacting for merely 15 min.

The plausible mechanism of the small molecule reaction (Fig. 1a)[28] implied that the stoichiometric balance between the key intermediate **E** and monomer **2** should be more crucial than the apparent monomer feed ratio for generating polymers with high molecular weights. However, monomer **1** is hard to be completely converted to **E** via step 1 and step 2. In this regard, excess monomer **1** is needed to promote the formation of **E** to meet the effective stoichiometry requirement between **E** and **2**. Indeed, the results shown in Table 1 (entries 1–5) and Supplementary Fig. 2A indicate that monomer ratios insert a great influence on the polymerizations. While the addition of an excess amount of internal diyne (**2a**) is unfavorable for the polymerization, the presence of excess 2-naphthol (**1a**) significantly improves the polymerization efficiency. The relative $M_n$ value of the polymeric product and the isolation yield gradually increase from 4100 (44%) to 12,200 (99%) as the monomer ratio of **1a** and **2a** increases from 1:1 to 4:1. The absolute $M_n$ values

**Fig. 1** Developing the spiroannulation reaction into a polyspiroannulation strategy. **a** Palladium(II)-catalyzed oxidative spiroannulations of naphthols and diarylacetylenes and the associated mechanism. **b** Palladium(II)-catalyzed polyspiroannulations of free naphthols and internal diynes for the construction of complex spiro-polymers in this work. [**1**]:[**2**] = molar feed ratio of monomer **1** and **2**.

**Table 1 Polymerization results of naphthols and internal diynes[a].**

| Entry | Monomers | [**1**]:[**2**] | Yield (%) | $M_n^b$ (MALLS)[c] | $M_w^b$ (MALLS)[c] | $Đ^b$ |
|---|---|---|---|---|---|---|
| 1 | **1a** + **2a** | 1:1 | 44 | 4100 (5800) | 5400 (6700) | 1.4 |
| 2 | **1a** + **2a** | 0.75:1 | 39 | 3600 (4200) | 4400 (4800) | 1.2 |
| 3 | **1a** + **2a** | 2:1 | 78 | 6700 (6900) | 10,600 (9100) | 1.6 |
| 4 | **1a** + **2a** | 3:1 | 82 | 8300 (7000) | 14,700 (9800) | 1.8 |
| 5 | **1a** + **2a** | 4:1 | 99 | 12,200 (10,000) | 21,300 (16,000) | 1.8 |
| 6 | **1a** + **2b** | 4:1 | 95 | 12,700 (7800) | 24,400 (13,000) | 1.9 |
| 7 | **1a** + **2c** | 4:1 | 42 | 9700 (18,000) | 14,300 (24,000) | 1.5 |
| 8 | **1a** + **2d** | 4:1 | 87 | 5100 (9000) | 7300 (12,000) | 1.5 |
| 9 | **1a** + **2e** | 4:1 | 87 | 5700 (9100) | 9900 (12,000) | 1.7 |
| 10 | **1b** + **2a** | 4:1 | 95 | 5400 (35,000) | 10,300 (39,000) | 1.9 |

[a]Carried out in DMSO under nitrogen at 120 °C for 24 h in the presence of Pd(OAc)$_2$, Cu(OAc)$_2$•H$_2$O and K$_2$CO$_3$. [**1**]:[**2**] = molar feed ratio of monomer **1** and **2**. [**2**] = 0.20 M, [Pd(OAc)$_2$] = 0.04 M, [Cu(OAc)$_2$•H$_2$O] = 0.84 M, [K$_2$CO$_3$] = 0.80 M
[b]Relative molecular weights determined by GPC in THF on the basis of a linear polystyrene calibration. $Đ$ = polydispersity = $M_w/M_n$
[c]Absolute molecular weights determined by GPC with a MALLS detector in DMF containing 0.1 M LiBr

determined by multiangle laser light scattering (MALLS) show a similar trend from 5800 to 10,000 (Supplementary Fig. 3). By isolation and analysis of the filtrate of the polymerization mixture at [**1**]:[**2**] = 4:1, the excess 2-naphthol is found to remain unreacted in a yield of 46% and transform into 1,1-bi-2-naphthol as the side product in a yield of 9% (Supplementary Figs. 4–11)[28].

On the basis of the optimized reaction conditions, we next performed the polymerizations of different monomer combinations to enrich the polymer structures and functionalities (Table 1, entries 6–10 and Supplementary Fig. 2B). Various internal diynes,

including monomer **2b**–**c** with different alkyl chain lengths and **2d**–**e** with the tetraphenylethene (TPE) luminophore, were employed for the polymerizations with 2-naphthol. The results show that all the tested diynes can afford polymers with high molecular weights (relative $M_n$ up to 12,700 and absolute $M_n$ up to 18,000) in moderate to good yields (up to 95%). The relatively lower reaction yield of P**1a/2c**–**e** is possibly due to the large steric hindrance of monomer **2c**–**e**, which hinders the smooth progress of this polymerization. The scope of monomer **1** was also investigated. When 2-naphthol was replaced with quinolin-6-ol

(**1b**) to polymerize with **2a**, P**1b**/**2a** with a high absolute $M_n$ of 35,000 was formed in a high yield of 95%.

**Structural characterization**. To assist the structural characterization and property investigation of the obtained polymers, model compound **4** was prepared through the coupling reaction of 2-naphthol and diphenylacetylene under conditions similar to those for polymerizations (Fig. 2a). To investigate the regioselectivity of this polymerization, we also conducted a model reaction using 2-naphthol and an asymmetrical internal diyne (**5**) as starting materials to generate model compound **6** (Supplementary Fig. 12). The structures of **4** and **6** were confirmed by high-resolution mass spectrometry (HRMS, Supplementary Figs. 13–14), FT-IR and NMR analysis.

The polymer structures were verified by comparing their characterization results with those of the corresponding monomers and model compounds (see Supplementary Methods for details). Taking P**1a**/**2a** as an example (sample taken from Table 1, entry 5), the [1]H NMR spectrum of P**1a**/**2a** shows no hydroxyl proton resonance of **1a** at a chemical shift ($\delta$) of 5.56 ppm. The signals related to the aromatic proton in b position of **1a** shift downfield to ~$\delta$ 6.16 ppm after polymerization (Fig. 2b–e and Supplementary Fig. 15). These observations indicate the occurrence of the polymerization. The comparison between the [1]H NMR spectrum of **6** and P**1a**/**2a** suggests that different isomeric units exist in the polymer structure (Supplementary Fig. 16). The [13]C NMR analysis provides more detailed information on the polymer structure. As shown in Fig. 2f–i, the resonance peaks assigned to the acetylenic carbon atoms at $\delta$ 89.71 and 88.20 ppm and the carbon atom in d position at $\delta$ 153.86 ppm all disappear in the spectrum of P**1a**/**2a**. Instead, a new peak at $\delta$ 196.23 ppm is observed due to the formation of carbonyl group in the polymer structure. Meanwhile, the aromatic carbon atom in e position of **1a** ($\delta$ 109.79 ppm)

is transformed to a quaternary carbon atom (f position, $\delta$ 76.16 ppm) after polymerization. The FT-IR and NMR spectra of the polymers largely resemble those of the model compounds (Supplementary Fig. 17). All these characterization results demonstrate that we have indeed obtained polymers with the desired spirocyclic structures as shown in Fig. 1b.

In addition, the structural characterization data of P**1a**/**2a** generated at different monomer ratios was also carefully analyzed. As illustrated in Fig. 2j and Supplementary Figs. 18–19, the signals associated with the C≡C group are obviously observed in the IR and [13]C NMR spectra of the polymeric products obtained at [**1a**]:[**2a**] = 0.75:1 or 1:1. Besides, the peaks related to the resonances of the protons of monomer **2a** at $\delta$ 4.00 ppm and $\delta$ 7.52–7.45 ppm are detected in the [1]H NMR spectra of these polymers (Supplementary Fig. 20). These results imply the presence of triple-bond containing species such as the unreacted internal diynes or end groups in the oligomeric products. However, the aforementioned spectral signals become weaker and weaker as the monomer feed ratio of **1a** and **2a** gradually increases and almost disappear at a ratio of 4:1. This tendency suggests that the presence of excess 2-naphthol can improve the polymerization efficiency by promoting the complete consumption of the diyne monomer, which is consistent with the abovementioned reaction mechanism.

**Preparation and post-polymerization of telechelic polymers**. Telechelic polymers are polymeric molecules capable of undergoing further polymerizations or other reactions through their reactive end groups[32,33]. They can serve as cross-linkers, chain extenders, and precursors for the construction of block and graft copolymers, star, hyperbranched or dendritic polymers, etc[34]. Herein we present an efficient step-growth polymerization strategy to prepare telechelic spiro-polymers by the C–H-activated polyspiroannulations based on their

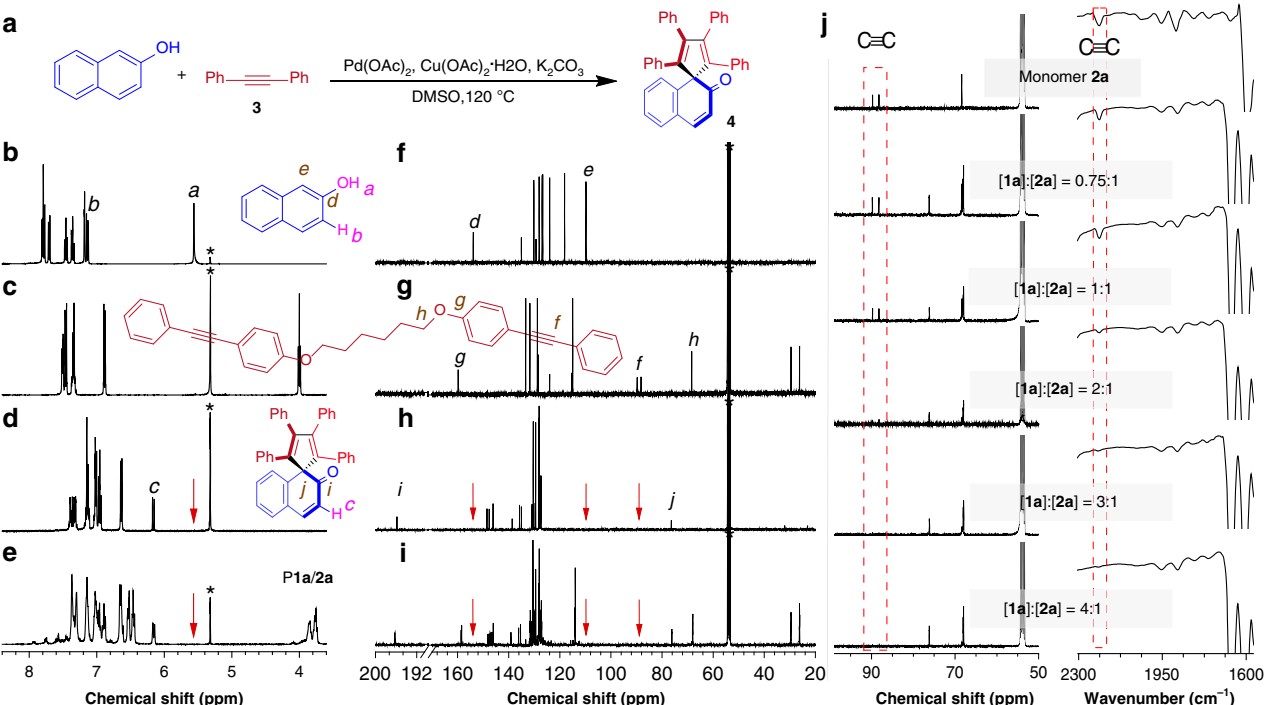

**Fig. 2** Structural characterization and analysis of the polymeric products. **a** Synthetic route to model compound **4**. **b–e** [1]H NMR spectra of **b** monomer **1a**, **c** monomer **2a**, **d** model compound **4**, and **e** P**1a**/**2a** in CD$_2$Cl$_2$. **f–i** [13]C NMR spectra of **f** monomer **1a**, **g** monomer **2a**, **h** model compound **4**, and **i** P**1a**/**2a** in CD$_2$Cl$_2$. **j** [13]C NMR spectra in CD$_2$Cl$_2$ (left) and IR spectra (right) of monomer **2a** and P**1a**/**2a** generated from polymerizations at different monomer ratios (sample taken from Table 1, entries 1–5, respectively).

apparent monomer-nonstoichiometry-promoted effect. This synthetic strategy should be more controllable and feasible than the traditional step-growth polymerizations[35,36].

As shown in Fig. 3, a phenylethynyl-terminated P1a/2a oligomer with a $M_n$ of 3800 and a $M_w$ of 5900 was prepared at a monomer feed ratio of 1:1. To test the reactivity of the triple-bond end groups, the isolated P1a/2a oligomer was used as a macromonomer to further react with 2-naphthol under the same palladium-catalyzed polymerization conditions (Fig. 3, route I). Delightfully, a polymer with a much higher molecular weight ($M_n = 9600$; $M_w = 24,100$) was generated in a high yield, indicating that the triple-bond end groups in the pre-polymer P1a/2a are still present. The characterization results shown in Supplementary Figs. 21–25 clearly demonstrate the consumption of triple-bond groups of the telechelic polymer to form macromolecules with extended polymer chains. Encouraged by this result, we then tested the possibility of utilizing the telechelic polymer to incorporate other functional units into the polymer backbones. As depicted in Fig. 3, 6-methoxy-2-naphthol (1c, route II) and 6-benzoyl-2-naphthol (1d, route III) were employed as co-monomers to react with telechelic P1a/2a. After polymerizations, P1ac/2a and P1ad/2a with much higher molecular weights than that of the precursor were successfully obtained in good to high yields. Their structures were confirmed by standard spectroscopic techniques (Supplementary Figs. 26–31). These results indicated that a telechelic spiro-polymer system that can function as a chain extender was successfully developed based on the monomer-nonstoichiometry-promoted effect of the present polymerization. It could also be possible to use the present telechelic polymer as a macromonomer in other alkyne-based polymerizations to allow the preparation of a broad range of functional materials with desirable physical features for diverse applications.

**Thermal properties**. The thermogravimetric analysis and differential scanning calorimetry results suggest that the obtained polymers possess high thermal stability and good morphological

stability possibly due to their rigid spirocyclic structures. As shown in Supplementary Fig. 32a, P1a–b/2a–e lose 5% of their weights at high temperatures of 379–451 °C. The TPE-containing polymers (P1a/2d–e) further exhibit a high char yield of 52% and 65%, respectively, after being heated to 800 °C. The glass transition temperatures of these spiro-polymers locate in the range of 125–279 °C (Supplementary Fig. 32b). In addition, all the obtained polymers show good solubility in common organic solvents, such as tetrahydrofuran (THF), dichloromethane (DCM), chloroform, 1,2-dichloroethane, etc., and can be readily fabricated into uniform thin films by simple spin-coating process. The excellent thermal properties and good solution processability allow these polymers to be used as potential heat-resistant coating materials.

**Photophysical properties**. 1,2,3,4-Tetraphenyl-1,3-cyclopentadiene (TPC) is a typical aggregation-induced emission luminogen (AIEgen) with efficient solid-state fluorescence[37,38]. Attracted by the unique TPC-containing structures of the model compound and polymers, we then investigated their photophysical properties. Their onset absorption wavelength ($\lambda_{onset}$) falls in the range of 382–433 nm (Supplementary Fig. 33). Interestingly, although 4 and P1a–b/2a–c possess TPC moiety in their structures, their solutions and powder show almost no fluorescence (Fig. 4a, b and Supplementary Fig. 34). We speculated that this phenomenon may result from the photoinduced electron transfer (PET) process. To assist the mechanistic study, model compounds 7 and P7 were synthesized by the reduction reaction of 4 and P1a/2a, respectively. Their structures were fully characterized and confirmed by GPC, FT-IR, $^1$H NMR, $^{13}$C NMR, and HRMS analysis (Supplementary Figs. 35–39). As shown in Fig. 4a, b, once the C=C bond adjacent to the carbonyl group was transformed to C–C bond, the resulting products emitted strong blue fluorescence under UV illumination. Due to the complexity of polymers, the frontier molecular orbitals of 4 and 7 were calculated instead by density functional theory to provide more hints on the photophysical properties of the polymers. The results shown in

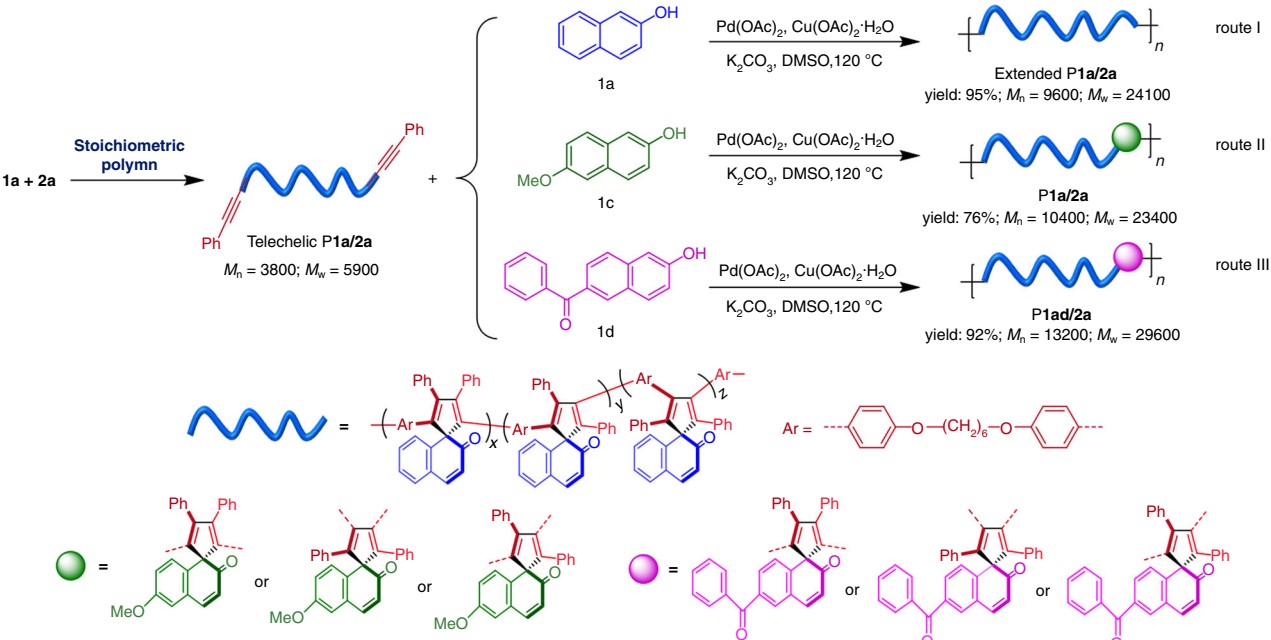

**Fig. 3** Preparation and post-polymerization of telechelic polymer. The telechelic polymer P1a/2a is prepared at a monomer feed ratio of 1:1, and the polymerizations of telechelic P1a/2a with 1a (route I), 1c (route II), and 1d (route III) are conducted based on the apparent monomer-nonstoichiometry-promoted effect.

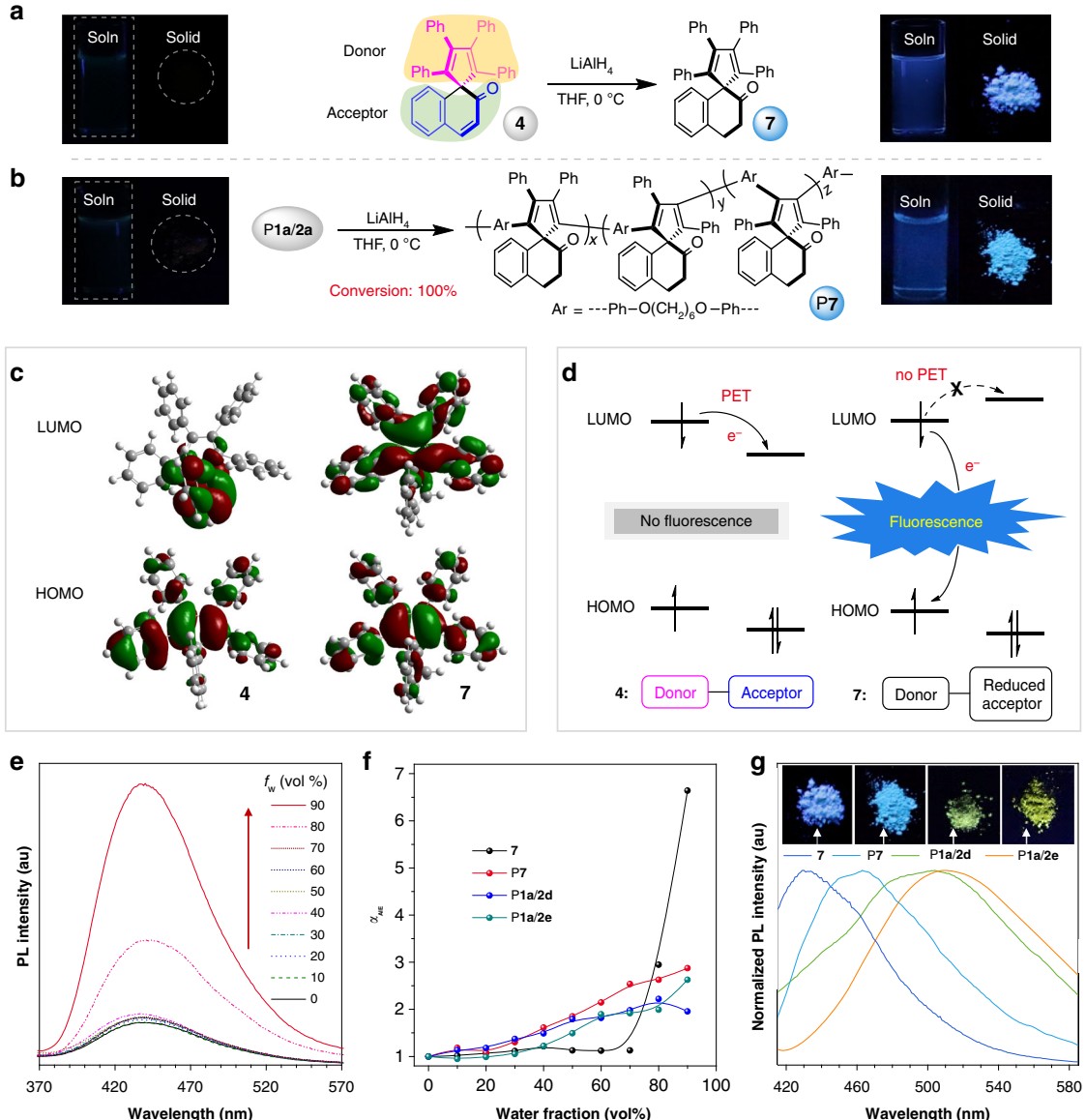

**Fig. 4** Photophysical properties of model compounds and polymers. **a** Reduction of model compound **4**. **b** Reduction of P**1a**/**2a**. Inset: fluorescent photographs of THF solutions and powder of **4** and P**1a**/**2a** (left side) and **7** and P**7** (right side). **c** Molecular orbitals of **4** and **7** in the ground state calculated by B3LYP/6-31G(d,p). **d** Schematic illustration of the modulation of fluorescence properties by the photoinduced electron transfer (PET) process. **e** Photoluminescence (PL) spectra of **7** in THF and THF/water mixtures with different water fractions ($f_w$). **f** Plot of the relative PL intensity ($I/I_0$) versus the composition of the aqueous mixtures of **7**, P**7**, P**1a**/**2d**, and P**1a**/**2e**. $\alpha_{AIE} = I/I_0$, where $I_0$ = intensity at $f_w = 0\%$. Solution concentration: 10 μM. **g** Normalized PL spectra of the powder of **7**, P**7**, P**1a**/**2d**, and P**1a**/**2e** and their associated fluorescent photographs. Excitation wavelength: 320 nm (for **7** and P**7**); 350 nm (for P**1a**/**2d** and P**1a**/**2e**). All fluorescent photographs were taken under UV irradiation at 365 nm.

Fig. 4c suggest that **4** possesses a donor–acceptor structure. The highest occupied molecular orbital (HOMO) of **4** is primarily localized on the TPC moiety, while its lowest unoccupied molecular orbital (LUMO) is mainly concentrated on the naphthalen-2(1*H*)-one part. In contrast, the electron cloud distribution of the HOMO and LUMO of the reduced product (**7**) is similar, indicative of the absence of donor–acceptor structure. Figure 4d provides a schematic illustration of the possible PET process. In model compound **4**, the excited electron is easily transferred from donor to acceptor, which hinders the return of electron to the HOMO of the donor and thus leads to fluorescence quenching[39]. After the reduction reaction, the HOMO of the reduced acceptor locates at a higher energy level than that of the donor to block the PET process. Therefore, the excited electron is allowed to undergo radiative decay to result in strong fluorescence in **7** and P**7**.

The reduced products (**7** and P**7**) were demonstrated to show an aggregation-enhanced emission (AEE) property (Fig. 4e, f and Supplementary Fig. 40a). The fluorescence quantum yields ($\Phi_F$) of the solid powder of **7** ($\Phi_F = 6.9\%$) and P**7** ($\Phi_F = 6.8\%$) were much higher than those of their THF solutions ($\Phi_F = 0.5\%$ and 2.8% for **7** and P**7**, respectively). These results indicate that the present polymerization route can in situ generate a pro-AIEgen system, which can become AIE-active under certain conditions. In addition to the reduced products, polymers P**1a**/**2d** and P**1a**/**2e** containing the typical AIEgen (TPE) were also found to exhibit AEE characteristics (Fig. 4f and Supplementary Fig. 40b–c). The AEE curve of compound **7** showed a turning point followed by a sharp increase, while the curves of polymers showed a slow variation trending. This difference may result from the more hydrophobic nature and more rigid structure of polymers, which

make them easier to form aggregates in aqueous mixtures with low water fractions ($f_w$). Taking P**1a/2e** as an example, the particle size analysis demonstrated the formation of nanoparticles with an average diameter of 185, 322, and 306 nm at low $f_w$ of 10%, 30%, and 50%, respectively. By contrast, model compound **7** can form large aggregates only at high $f_w$. Nanoparticles with average diameters of 362 and 424 nm were detected in its 80% and 90% aqueous mixtures, respectively. The AEE effect of these polymers can be ascribed to the restriction of intramolecular motions by the physical constraint of aggregates[40]. The fluorescence color of **7**, P**7**, P**1a/2d**, and P**1a/2e** powder covers the range from deep-blue to yellow with the maximum emission wavelength ranging from 430 to 510 nm (Fig. 4g). The efficient solid-state emission enables these polymers to find potential applications as light-emitting coating materials and fluorescent probes[41].

**Refractive index and chromatic dispersion**. High-refractive-index polymers with small chromic dispersion are desirable materials for the fabrication of advanced optoelectronics due to their advantages of good processability, low cost, light weight, excellent impact resistance and dyability over inorganic optical materials[42]. As shown in Fig. 5a and Supplementary Table 6 (entries 1–6), all the obtained polymers show high refractive index ($n$) ranging from 1.9282 to 1.5710 in a wide wavelength region of 370–1700 nm due to the presence of numerous polarized phenyl rings in the polymer structures[43]. Their $n$ values at 632.8 nm ($n_{632.8}$) are in the range of 1.6407–1.6896, which are much higher than those of the commercially important optical

plastics ($n = \sim$1.50–1.60)[44]. Meanwhile, these spiro-polymers show small chromic dispersions in both the visible region ($D = $ 0.071–0.100) and the IR region ($D' = $ 0.008–0.028).

Moreover, the refractive properties of the obtained polymers are responsive to UV irradiation. As indicated in Fig. 5b and Supplementary Table 6 (entries 1 and 7–10), the $n$ values of P**1a/2a** gradually decreased with an increase in UV exposure time. The $n$ value at 1550 nm ($n_{1550}$), a wavelength of telecommunication importance, decreased by 0.0147, 0.0245, 0.0273, and 0.0368 after UV irradiation for 10, 20, 30, and 40 min, respectively. Upon UV exposure for 40 min, the $n$ values of P**1a/2a** at the same wavelength region significantly dropped from 1.8096–1.6150 to 1.7128–1.5783. The results shown in Supplementary Figs. 41–43 suggest that such a large change in refractive index is possibly due to the UV irradiation-induced structure change of the polymers. There are multiple reactive sites in the polymer structures that could be photo-oxidized under the irradiation of strong UV light, which will reduce the electronic conjugation and thus lead to the decrease in their refractive indices. The excellent tunability of the film refractivity by facile UV irradiation enables the present polymers to find potential applications in optical data storage devices, gradient-index optics and integrated photonics technology.

**Photopatterning**. The generation of complex micro- and nano-patterns on polymeric surfaces or thin films is important for the development of biological sensing systems, optical writing and reading, anti-counterfeiting applications, and the construction of optical display devices[45–47]. Considering the good film-forming ability and photosensitivity of the obtained polymers, we thus explore their potential applications in photopatterning. Smooth polymer thin films were first fabricated by spin-coating their 1,2-dichloroethane solutions on silicon wafers. By simply exposing these thin films to UV light in air for 20 min at room temperature through a negative copper photomask (Fig. 6a), well-resolved two-dimensional fluorescent patterns with both positive and negative image readout capability can be readily generated and clearly visualized.

The thin films of P**7**, P**1a/2d**, and P**1a/2e** show obvious light emission due to their AEE features. After the photolithography process, the unexposed square parts of the thin films remain blue or green emissive while the emission of the exposed lines is quenched to give a clear turn-off-type (positive) 2D photopattern. The detected intensity profile indicates that the contrast between the bright and dark state is more than 3.8-fold (Fig. 6b and Supplementary Fig. 44a–d). Different from the AEE-active polymers, P**1a/2a** and P**1b/2a** show an interesting phenomenon of UV-activated fluorescence. As depicted in Fig. 6c and Supplementary Fig. 44e–f, well-resolved turn-on-type (negative) 2D photopatterns with luminescent lines and non-emissive squares are generated with more than 3.1-fold contrast. It is worth noting that these patterns can also be clearly observed under room light (Fig. 6d and Supplementary Fig. 45) possibly due to the remarkable change in the thin film refractivity upon UV irradiation. In addition to grid pattern, other patterns with different sizes and contents can also be readily generated and clearly visualized. As shown in Fig. 6e and Supplementary Fig. 46, different types of flower patterns are fabricated using the same photomask based on the photobleaching and photoactivation process of P**7** and P**1a/2a**, respectively. These patterns are stable and the detected contrasts remain constant even after storage under ambient conditions/normal room light for more than 6 months.

Why these polymers can show photoresponsive fluorescence? The possibility of photo-crosslinking process can be excluded

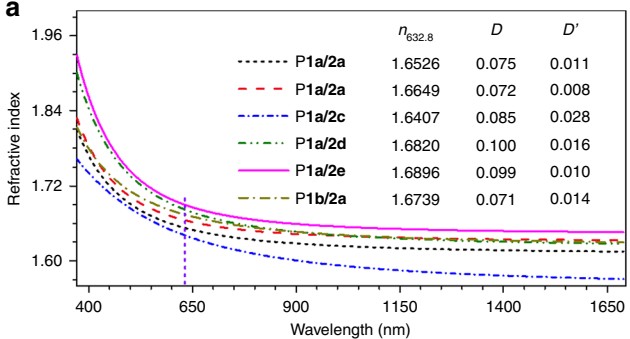

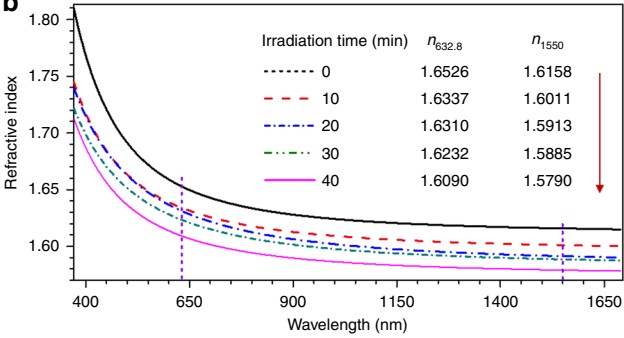

**Fig. 5** Refractive index and chromatic dispersion of polymer thin films. **a** Wavelength dependence of refractive indices of thin films of P**1a–b/2a–e**. **b** Change in the refractive index of a thin film of P**1a/2a** by UV irradiation for different durations. Abbreviation: $n =$ refractive index; $D =$ chromatic dispersion in the visible region $= (n_F - n_C)/(n_D - 1)$, where $n_D$, $n_F$, and $n_C$ are the $n$ values at wavelengths of Fraunhofer D, F, and C spectral lines of 589.2, 486.1, and 656.3 nm, respectively; $D' =$ chromatic dispersion in the IR region $= (n_{1064} - n_{1550})/(n_{1319} - 1)$, where $n_{1064}$, $n_{1319}$, and $n_{1550}$ are the $n$ values at 1064, 1319, and 1550 nm, respectively[57].

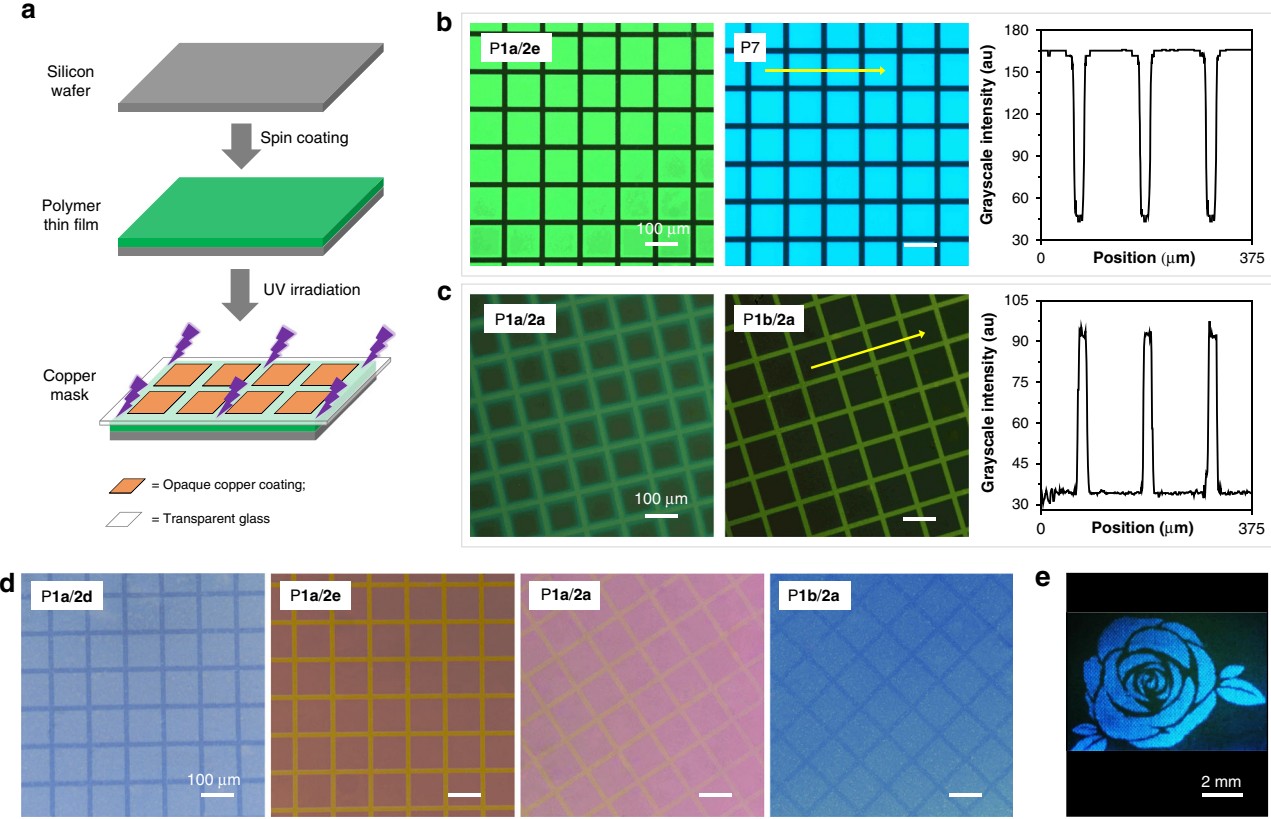

**Fig. 6** Fabrication of photopatterns using polymer thin films and the obtained images. **a** Schematic illustration of the fabrication of photopatterns: two-dimensional photopatterns were generated by the photo-masked UV irradiation of polymer thin films on silicon wafers. **b** Fluorescent images of the turn-off-type photopatterns (left) and the grayscale intensity profile of the arrowed area in the patterned P**7** film. **c** Fluorescent images of the turn-on-type photopatterns (left) and the grayscale intensity profile of the arrowed area in the patterned P**1b/2a** film. The fluorescent images in (**b**) and (**c**) are taken under 330–380 nm UV illumination using a fluorescent microscope. **d** Photographs of the photopatterns taken under normal room light using an optical microscope. All images in (**b**), (**c**), and (**d**) share the same scale bar = 100 μm. **e** A fluorescent flower-like photopattern of P**7** taken under UV irradiation at 365 nm using a camera.

because all the photopatterned films can be easily washed off without any residue by their good solvents such as THF and DCM. To assist the mechanistic study, the UV responsiveness of model compounds **4** and **7** was investigated. As shown in Supplementary Fig. 47, the PL intensity of **4** was obviously enhanced after UV irradiation whereas the fluorescence of **7** was significantly weakened by UV light, which is consistent with their corresponding polymers. The opposite photoresponsive behavior between P**1a/2a** (**4**) and P**7** (**7**) suggested that the carbonyl-activated C=C double bonds may play a crucial role in the UV-activated fluorescence of P**1a/2a** and P**1b/2a**. We then studied the UV-induced structural change by UV-vis absorption spectroscopy, $^1$H NMR and X-ray photoelectron spectroscopy, etc. The results and discussions in Supplementary Figs. 42–43 and 48–52 suggested that the UV-activated fluorescence of P**1a/2a** and P**1b/2a** may be due to the occurrence of complicated photo-oxidation reactions with the involvement of radical species to generate new structures with obvious solid-state fluorescence. As depicted in Supplementary Figs. 50–51, the resonance signal of the carbonyl-activated alkene proton of **4** (at ~δ 6.16) was obviously weakened after UV irradiation, and meanwhile some new resonance peaks were detected in the aromatic proton region of the $^1$H NMR spectra of both the irradiated **4** and **7**. These results suggested that the photochemical reactions of this complex spirocyclic structure should involve the carbonyl-activated C=C together with some other reactive sites. As discussed above, the poor solid-state fluorescence of **4** and

P**1a–b/2a–c** may result from the PET process. Therefore, the UV-activated fluorescence of P**1a/2a** and P**1b/2a** might be due to the block of the PET process by the photoreactions. For the AEE-active polymers, the results and discussions provided in Supplementary Figs. 47 and 51–53 implied that the UV-quenched fluorescence of P**1a/2d–e** and P**7** may result from the occurrence of photo-oxidative bleaching process of their AIE moieties.

**Applications in integrated silicon photonics.** Integrated silicon photonics has become a popular technology platform to enable applications in intra-datacenter optical interconnects, nonlinear and quantum photonics and lab-on-a-chip optical biosensing[48–51]. As a key building block in integrated silicon photonics, micrometer-sized ring (microring) resonators constitute a highly phase-sensitive optical device with narrow-band wavelength selectivity. However, a major hurdle for practical applications of silicon-based microrings is that the resonance wavelengths ($\lambda_{res}$) are highly sensitive to inevitable fabrication imperfections, and thus are difficult to be pre-designed. Although some active $\lambda_{res}$-tuning methods have been reported based on thermo-optic[52,53] or electro-optic[54] effects, these tuning methods require constantly applying an electrical voltage to either heat up an integrated electrode in close proximity to the microring or inject or deplete carriers in the case of silicon microrings. Here we demonstrate a simple one-time UV-irradiation method to permanently tune the

$\lambda_{res}$ of a polymer-clad microring by applying our photosensitive polymer as the upper-cladding layer.

Figure 7a shows schematically a silicon-based microring resonator evanescently coupled with a bus waveguide. The $\lambda_{res}$ of a microring is positively correlated with its waveguide effective refractive index ($n_{eff}$) according to the phase-matching condition (see Supplementary Methods for details). For a polymer-clad microring, its $n_{eff}$ depends on both the $n$ of the polymer cladding and the spatial overlap between the optical mode and the cladding[55,56]. As the $n$ values of P**1a**/**2a** are UV-tunable, the $n_{eff}$ and $\lambda_{res}$ of P**1a**/**2a**-coated microring devices are possible to be tuned by UV irradiation. To test this possibility, we then spin-coated P**1a**/**2a** on the pre-fabricated silicon nitride (Si$_3$N$_4$) microring resonators on a silicon chip (Fig. 7b). Figure 7c, d shows the finite-element method (FEM)-simulated mode-field amplitude distributions in the polymer-coated Si$_3$N$_4$ waveguide upon transverse-magnetic (TM) polarization (the electric field is perpendicular to the substrate) and transverse-electric (TE) polarization (the electric field is parallel to the substrate),

respectively. The simulation results suggest that the waveguide mode field spatially overlaps with the polymer layer. For TM polarization, the optical field has more spatial overlap with the top cladding, whereas the optical field for TE polarization shows more spatial overlap with the side cladding.

The transmission spectra of six P**1a**/**2a**-coated microring resonators were then measured in both TM and TE polarizations with different UV exposure durations under the same UV power. Figure 7e shows the obtained spectra of one device near 1550 nm wavelength in TM polarization with the UV exposure duration varying from 0 to 40 min at an interval of 10 min. As expected, a blue-shift of $\lambda_{res}$ occurred upon UV irradiation and a longer UV exposure duration results in a linearly increased blue-shift ($\Delta\lambda$) of $\lambda_{res}$. Figure 7f shows the extracted $|\Delta\lambda/\lambda_{res}|$ of all six measured devices in TM polarization. In all the cases, the $\lambda_{res}$ blue-shifts linearly as the exposure duration increases, which indicates that this strategy is applicable to a series of different devices. Figure 7g shows the comparison between the extracted $|\Delta\lambda/\lambda_{res}|$ of the microring devices in TM polarization and the extracted $|\Delta n/n|$ of

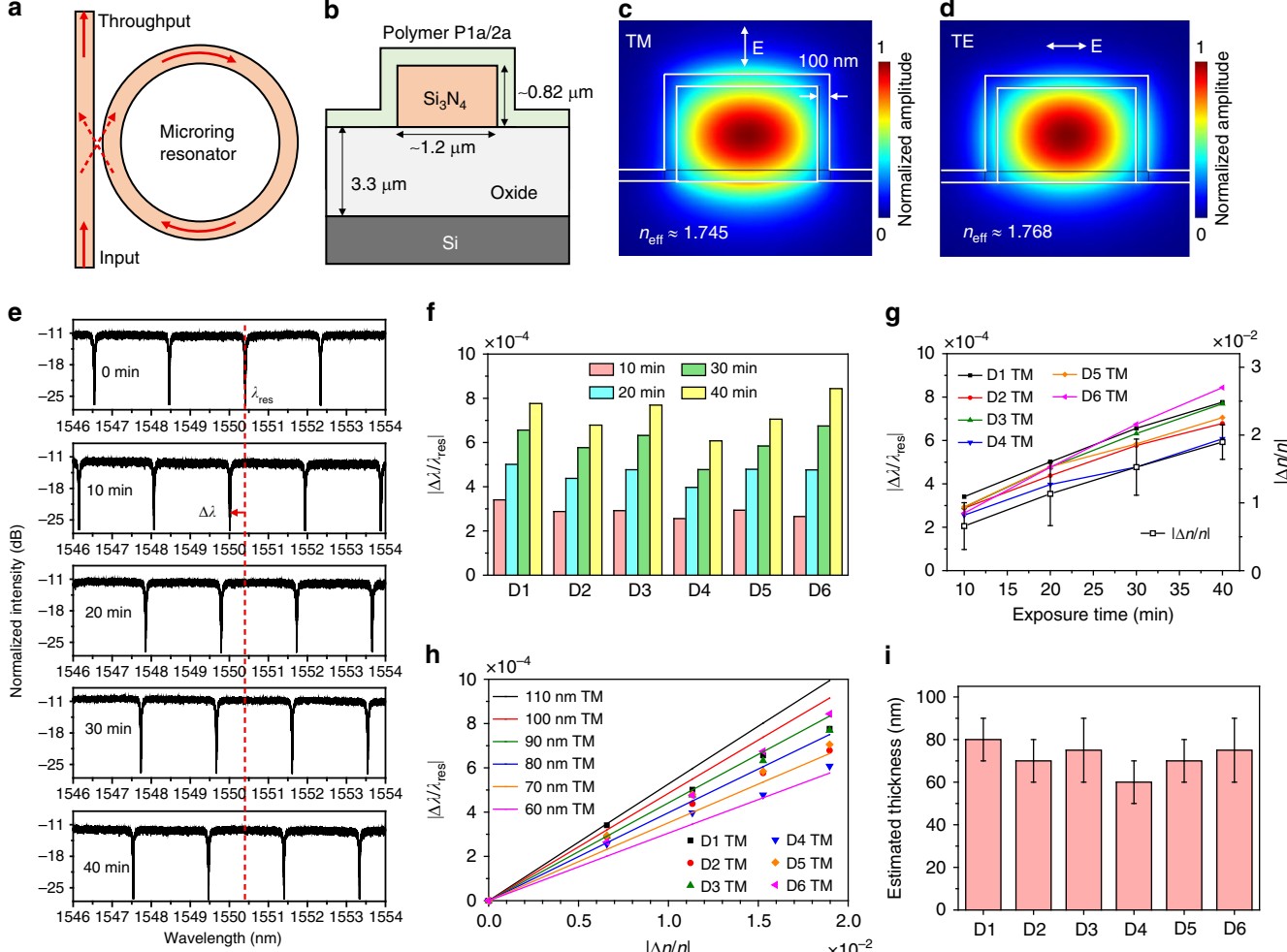

**Fig. 7 Applications of photoresponsive polymer films in integrated silicon photonics. a** Schematic of a silicon-based microring resonator evanescently coupled with a bus waveguide. **b** Schematic of the cross section of a polymer-coated Si$_3$N$_4$ waveguide. **c, d** FEM-simulated mode-field amplitude distributions with **c** TM polarization and **d** TE polarization. **e** Measured transmission spectra of a microring near 1550 nm wavelength in TM polarization with UV exposure duration varying from 0 to 40 min at an interval of 10 min. **f** Extracted $|\Delta\lambda/\lambda_{res}|$ of all the six measured devices in TM polarization. **g** Comparison between the extracted $|\Delta\lambda/\lambda_{res}|$ of the microrings in TM polarization and the extracted $|\Delta n/n|$ of the polymer films as a function of the UV exposure duration. The error bars indicate the standard deviations. **h** Comparison of $|\Delta\lambda/\lambda_{res}|$ between the six measured devices and the simulations in TM polarization with the polymer thickness varying from 60 to 110 nm at an interval of 10 nm. **i** Estimated polymer thickness of the six devices based on both TM and TE polarizations. The upper and lower limits of the error bars indicate the estimated thickness values from TM and TE polarizations, respectively. The height of each colored bar is defined as the average value of the upper and lower limit.

the polymer films as a function of the UV exposure duration. These results clearly indicate that the observed blue-shift of $\lambda_{res}$ is directly related to the UV-induced refractive index change of the polymer cladding and the $\Delta\lambda$ can be finely modulated by controlling the UV exposure duration at a given UV power. Similar conclusions can also be drawn from the results with TE polarization (Supplementary Figs. 54–56).

As illustrated in Fig. 7h and Supplementary Fig. 57, the polymer thickness coated on the devices was estimated by finding the closest match between the experimental data of $|\Delta\lambda/\lambda_{res}|$ of the six devices (symbols) and the FEM-simulated results with different polymer thickness (lines). For each device, two estimated polymer thickness values are obtained based on TM and TE polarizations. The results summarized in Supplementary Table 7 suggest that the polymer thickness on the waveguide top surface (estimated from TM polarizations) is generally thicker than the sidewall (estimated from TE polarizations). The different film thickness distribution in one device will affect the spatial overlap between optical modes and the claddings, which is why the $\lambda_{res}$ in TM polarization show more sensitive response than that in TE polarization. The small variation in the film thickness among these devices (Fig. 7i) can explain the slight difference in their sensitivity to UV irradiation. In short, this work provides a proof-of-concept study for the potential application of such photoresponsive polymer material in integrated silicon photonics and provides a simple UV-irradiation strategy to permanently tune the $\lambda_{res}$ of microring resonators.

## Discussion

In summary, we have developed a straightforward C–H-activated polyspiroannulation method that can transform the readily available and inexpensive 2-naphthols and diynes into valuable photoresponsive spiro-polymers with complex structures and multifunctionalities. The multisubstituted spirocyclic segments are generated in situ in the polymer backbones from the palladium-catalyzed polyspiroannulations based on the C–H activation chemistry, which is hard to be achieved by other polymerization methods. Unlike traditional homogeneous two-component polymerizations, one of the monomers in this step-wise polyspiroannulation is monofunctional and the presence of excess monofunctional monomers can significantly improve the polymerization efficiency. Taking advantage of its unique reaction mechanism, the present polyspiroannulation can be applied for the preparation and post-functionalization of spirocyclic tele-chelic polymers to further expand the libraries of functional spiro-polymers. The obtained polymers show high thermal and morphological stability and interesting photoresponsive optical properties due to their unique structures. Well-resolved two-dimensional fluorescent images with both turn-off and turn-on mode can be readily fabricated based on the photobleaching and photoactivation process of different polymer films under strong UV irradiation. Moreover, we utilized the photoresponsive refractive index of these polymer thin films to permanently modify the resonance wavelengths of microring resonators by UV irradiation. The applicability could be further strengthened by developing polymer materials with a larger refractive index change in response to UV irradiation or using polymer substrates to extend the application to flexible photonics. It is anticipated that this work not only can promote the further development of C–H-activated stepwise polymerizations of monofunctional monomers with diynes, but also can open up an avenue for the design and applications of photoresponsive polymers in integrated photonic or electrophotonic devices, which will benefit the progress of both the photoresponsive polymers and the integrated silicon photonics technique.

## Methods

**Polymer synthesis**. All the polymerization reactions were carried out under nitrogen using the standard Schlenk technique. A typical polymerization procedure for the preparation of P**1a**/**2a** (Table 1, entry 5) is given below as an example. Into a 10 mL Schlenk tube with a stirring bar was added 2-naphthol (115.3 mg, 0.8 mmol), internal diyne **2a** (94.1 mg, 0.2 mmol), Pd(OAc)$_2$ (9.0 mg, 0.04 mmol), Cu (OAc)$_2\cdot$H$_2$O (167.7 mg, 0.84 mmol), and K$_2$CO$_3$ (110.6 mg, 0.8 mmol) in 1 mL DMSO. The reaction mixture was stirred under nitrogen in a sealed Schlenk tube at 120 °C for 24 h and then cooled to room temperature. To remove the catalytic species, especially the copper salt, the resulting mixture was first dissolved with THF and centrifuged for several times. Then the supernatant solution was passed through a simple column filled with neutral Al$_2$O$_3$ powder and added dropwise to 160-mL hexane/chloroform mixture (7:1 v/v) under vigorous stirring. The precipitates were collected by filtration, and then washed with hexane and dried in vacuum at room temperature to a constant weight.

**Reporting summary**. Further information on research design is available in the Nature Research Reporting Summary linked to this article.

## Data availability

The authors declare that the all the data supporting the findings of this study are available within this article and its Supplementary Information files, and are also available from the corresponding authors upon reasonable request.

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

## Acknowledgements

This work was partially supported by the National Science Foundation of China (21490570 and 21490574), the University Grants Committee of Hong Kong (AoE/P-03/08), the Research Grants Council of Hong Kong (16308116 and 16305618), the Innovation and Technology Commission (ITC-CNERC14S01 and ITCPD/17-9), and the Science and Technology Plan of Shenzhen (JCYJ20160229205601482 and JCYJ20160428150429072).

## Author contributions

B.Z.T. conceived and designed the experiments. T.H. performed all the synthesis, structural characterization, photophysical, and light refractivity measurements and fabricated the photopatterns. A.W.P. and Z.Y. performed the experiments and data analysis on microring resonators. Z.Q., Z.Z., and J.W. analyzed the data and participated in the discussion. K.W fabricated the microring resonators. J.W.Y.L. and A.W.P. revised the paper. H.T., Z.Y., J.W.Y.L., and B.Z.T. wrote the paper with comments from all authors.

## Competing interests

The authors declare no competing interests.
