## [Peer Review File · Nature Communications]

Reviewers' comments:

Reviewer #1 (Remarks to the Author):

This manuscript reports the synthesis of spiro-containing photoreponsive polymers based on Pd catalyzed reaction of dialkyne compounds with naphthols. It is demonstrated that moderate molar mass materials are obtained that exhibit good thermal properties and processability. Furthermore, the materials exhibit very interesting photophysical properties as it demonstrated for high RI materials, photopatterning as well as tunable RI materials for coating of ring resonators. As this is a very creative and very novel of making such complex polymers, in combination with the excellent properties of the obtained materials, I support publication of this work in Nature Communications after the following comments have been addressed:

Detailed comments:

1. In my opinion there is no need to overemphasize the nonstoichiometric nature of the process. The process is very elegant on its own and for me this continuous repetition of the nonstoichiometric nature is not adding value, but actually feel like the authors are trying to oversell a not so important phenomenon. This should only be mentioned few times and should not be in the title. Instead, the title should include the spiroopyran containing polymers as this is the important aspect of the work.
2. In the abstract, the sentence 'The presence of excess... efficiency' should be removed as no details are given on the polymerization process making this sentence redundant. Some more details on the polymerization process would be welcomed in the abstract instead.
3. The term photomodulated should be replaced by photoresponsive, which is a more commonly used term
4. P7: In the discussion it would be better to discuss Mn rather than Mw. Furthermore, the absolute molar mass of some of the polymers should be reported as the molar mass of these rigid structures is often overestimated by SEC relative to flexible polymer standards. It may well be that the 'polymers' are more of oligomeric nature. The authors should include SEC with light scattering detection, or MALDI-TOF data (if they get lucky with the latter). For the telechelic polymers, the Mn can also be calculated by Mn based on the polymer end-groups, potentially with quantitative ¹³C NMR using the alkyne C-atoms.
Overlay of the SEC traces of the polymers reported in Table 1 should also be included in the supporting info.
5. P11: The polymer end-groups are not living. Instead it can be mentioned that they are still present
6. P16: Why does the RI increase upon photoirradiation? This should be further discussed and demonstrated.
7. P18: Proof that the photobleaching is due to photo-oxidative bleaching should be preferably provided, rather than the references to other polymer structures.
8. P18: Why do P1a/2a and P1b/2a show UV-activated fluorescence? This has not been addressed and an explanation should be provided.
9. P22: Replace 'Discussion' with 'Conclusions'

Reviewer: Richard Hoogenboom

Reviewer #2 (Remarks to the Author):

Tang and co-workers report on a new step-growth polymerization for the synthesis of photomodulatable polymers. The work appears to have been carefully performed however the novelty and impact of the work does not seem best suited for Nature Communications. This work seems better suited for a polymer specific journal.

The authors claim term a monomer nonstoichiometric strategy, but this term is somewhat misleading. Although the authors use a nonstoichiometric quantity of monomers, this optimization is mostly to match the catalyst reactivity for the synthesis of higher molecular weight polymers. This reviewer suggests using different terminology to describe their approach.

The authors then characterize the polymers and their optical properties. Although many interesting results are reported, the photomodulatable properties are reported as a broad collection of highlights of results that this reviewer believes would be better presented if they were presented more thoroughly in a series of separate manuscripts.

Reviewer #3 (Remarks to the Author):

The authors reported the synthesis of a series of photomodulatable spirocyclic polymers using nonstoichiometric stepwise polymerization and investigated their optical and thermal properties. They also use these polymers to fabricate fluorescent photopatterns and make integrated silicon photonics. A lot of data and interesting results were provided, but the corresponding discussions were not enough. This paper might be suitable for publication in this high-level journal after the following issues were settled.

More comments:

- (1) In Figure 4f, the curve of compound 7 showed a turning point and following sharp increase, while the other polymers only showed a slow variation trending. Please explore the possible reason.
- (2) On line 346 (page 16), they ascribed the change in refractive properties to the photosensitive vinyl and carbonyl groups. Please indicate the possible photo-reaction of these polymers under UV irradiation.
- (3) In Figure 6c, P1a/2a and P1b/2a show an interesting phenomenon of UV-activated fluorescence. Please explain this unusual phenomenon and the possible UV-induced structural change.
- (4) Please carefully check the typos in this manuscript, for example, on line 350, "the n value at 1500 nm (n1550)".

Reply to Reviewers' Comments for Manuscript NCOMMS-19-15699A

Response to the Comments of Reviewer 1

Comments: This manuscript reports the synthesis of spiro-containing photoreponsive polymers based on Pd catalyzed reaction of dialkyne compounds with naphthols. It is demonstrated that moderate molar mass materials are obtained that exhibit good thermal properties and processability. Furthermore, the materials exhibit very interesting photophysical properties as it demonstrated for high RI materials, photopatterning as well as tunable RI materials for coating of ring resonators.

As this is a very creative and very novel of making such complex polymers, in combination with the excellent properties of the obtained materials, I support publication of this work in Nature Communications after the following comments have been addressed:

Response: We greatly appreciate the positive comments from the reviewer on our work. We have carefully read each of the comments and tried our best to address them. Our point-by-point response is given below.

(1) In my opinion there is no need to overemphasize the nonstoichiometric nature of the process. The process is very elegant on its own and for me this continuous repetition of the nonstoichiometric nature is not adding value, but actually feel like the authors are trying to oversell a not so important phenomenon. This should only be mentioned few times and should not be in the title. Instead, the title should include the spiro-pyran containing polymers as this is the important aspect of the work.

Response: Thanks for the reviewer's constructive suggestions. Accordingly, we have reduced the frequency of mentioning the nonstoichiometric nature of the polymerization in the revised manuscript. Meanwhile, the importance of this polymerization process itself and the unique polymer structure with spirocyclic units are highlighted. The title was revised as "photoresponsive spiro-polymers generated in situ by C-H-activated polyspiroannulation". The changes we made in the revised manuscript and supporting information are highlighted in red for easy identification.

(2) In the abstract, the sentence 'The presence of exces... efficiency' should be removed as no details are given on the polymerization process making this sentence redundant. Some more details on the polymerization process would be welcomed in the abstract instead.

Response: Thanks for the reviewer's suggestion. We have corrected the abstract accordingly in the revised manuscript. The sentence 'The presence of exces... efficiency' was removed and some more details on the polymerization process were described in the revised abstract.

(3) The term photomodulated should be replaced by photoresponsive, which is a more commonly used term.

Response: Thanks for the reviewer's suggestion. We have corrected it accordingly in the revised manuscript.

(4) P7: In the discussion it would be better to discuss M_n rather than M_w . Furthermore, the absolute molar mass of some of the polymers should be reported as the molar mass of these rigid structures is often overestimated by SEC relative to flexible polymer standards. It may well be that the 'polymers' are more of oligomeric nature. The authors should include SEC with light scattering detection, or MALDI-TOF data (if they get lucky with the latter). For the telechelic polymers, the M_n can also be calculated by M_n based on the polymer end-groups, potentially with quantitative ^{13}C NMR using the alkyne C-atoms.

Overlay of the SEC traces of the polymers reported in Table 1 should also be included in the supporting info.

Response: Thanks for the reviewer's suggestions.

1) Following the reviewer's suggestion, we have revised the discussion in the manuscript to take the number-average molecular weights into account. The changes we made are highlighted in red in the revised manuscript for easy identification.

2) The absolute molar mass of the polymers shown in Table 1 and Figure 3 were measured by SEC with a multiangle laser light scattering (MALLS). The corresponding results and discussions were provided in the revised manuscript and supporting information (Table 1, Supplementary Fig. 3 and Supplementary Fig. 25). In the revised manuscript, the relative molecular weights determined by THF SEC calibrated using polystyrene standards were mainly used when we discuss the optimization process of the polymerization conditions. The absolute molecular weights determined by MALLS-SEC were provided to better characterize the polymers obtained using different monomer ratios or different monomer combinations. As shown in the revised Table 1, most of the absolute M_n values determined by MALLS-SEC are similar to or even higher than their corresponding relative M_n values determined by calibrated SEC. The absolute M_n values of **P1a–b/2a–d** are in the range of 7800–35000, which is indicative of the polymeric nature of the obtained products.

Also, we have obtained the MALDI-TOF-MS spectra of some of the polymers by a Bruker autoflex III mass spectrometer with DCTB matrix in the linear mode (as shown in Figure R1-R4).

Figure R1. MALDI-TOF mass spectrum of **P1a/2a** obtained at a monomer ratio of $[1a]:[2a] = 1:1$ (sample taken from Table 1, entry 1).

Figure R2. MALDI-TOF mass spectrum of **P1a/2a** obtained at a monomer ratio of $[1a]:[2a] = 0.75:1$ (sample taken from Table 1, entry 2).

Figure R3. MALDI-TOF mass spectrum of **P1a/2a** obtained at a monomer ratio of **[1a]:[2a] = 2:1** (sample taken from Table 1, entry 3).

Figure R4. MALDI-TOF mass spectrum of **P1a/2d** (sample taken from Table 1, entry 8).

The M_n of the telechelic polymer (i.e. **P1a/2a** obtained at a monomer ratio of 1:1) was determined by MALLS-GPC in the revised manuscript. We have also tried to calculate the M_n of the telechelic polymer by the quantitative ^{13}C NMR approach. The experiment was carried out on a Bruker AV 400 NMR spectrometer using polymer solution containing 0.05 mol/L chromium(III) acetylacetonate relaxation agent in CDCl_3 . The spectrometer is run using a 90° pulse τ of 7.7 μs , a spectral width of 37 kHz, a relaxation delay d1 of 7 s, an acquisition time of 0.87 s and inverse-gated decoupling. The results were shown in Figure R5 and Table R1. For end-group

integration, the two signals of C≡C end-group were added and then divided by two. From these data, the M_n of telechelic **P1a/2a** is calculated to be 3021, which is similar to the result measured by GPC ($M_n = 3800$).

Figure R5. Quantitative ^{13}C NMR spectrum of telechelic **P1a/2a** recorded in CDCl_3 with 0.05 mol/L $\text{Cr}(\text{acac})_3$.

Table R1. Deconvolution results for the quantitative ^{13}C NMR spectrum of telechelic **P1a/2a**.

Peak chemical shift (δ ppm)	Peak Integral	Peak Assignment
196.04	37.38	C=O backbone
89.22	9.01	C≡C end-group
87.74	8.95	C≡C end-group

3) Overlay of the SEC traces of the polymers reported in Table 1 were provided in **Supplementary Fig. 2** of the revised supporting information.

Supplementary Fig. 2 Overlay of the GPC traces of the polymers reported in (A) Table 1, entries 1–5 and (B) Table 1, entries 5–10 (measured by GPC in THF on the basis of a linear polystyrene calibration).

(5) P11: The polymer end-groups are not living. Instead it can be mentioned that they are still present.

Response: Thanks for the reviewer's suggestion. We have corrected the description accordingly in the revised manuscript.

(6) P16: Why does the RI increase upon photoirradiation? This should be further discussed and demonstrated.

Response: Thanks for the reviewer's question and suggestion. As described in the manuscript, the RI of the polymer gradually decreased with an increase in photoirradiation time. This phenomenon is possibly due to the structural change induced by photoirradiation. It is known that RI can be influenced by the polarizability of the components composing the polymer repeating unit and the presence of highly polarizable π -conjugated functionalities are beneficial for increasing the refractive indices of polymers (*J. Mater. Chem.*, 2009, 19, 8907). Therefore, the decrease in electronic conjugation could result in the decrease in RI. For example, we have measured the RI of P7 and compared it with that of P1a/2a (Supplementary Table 6 and Supplementary Fig. 41) in the revised supporting information. The results showed that the RI of P7 is lower than that of P1a/2a, which indicated that the conversion of the unsaturated C=C to saturated C-C can lead to a decrease in RI due to the decreased electronic conjugation. There are multiple reactive sites in the polymer structures that could be photo-oxidized, which will reduce the electronic conjugation and thus lead to the decrease in RI.

Supplementary Fig. 41 Wavelength dependence of refractive indices of thin films of P1a/2a and P7.

The photoirradiation process was conducted in air using a strong UV light source (the incident light intensity was $\sim 18.5 \text{ mW cm}^{-2}$ and the applied power of the Mercury Arc Lamp was 180 W). Under the strong UV irradiation, the multi-substituted spirocyclic unit in the polymers are possible to undergo some photochemical reactions. The absorption spectrum of P1a/2a were measured before

and after UV irradiation. The results shown in Supplementary Fig. 42A suggested that the strong UV irradiation could indeed lead to a decrease in the electronic conjugation of **P1a/2a**. Due to the intrinsic complexity and polydispersity of polymers, the UV responsiveness of model compound **4** was investigated to assist the mechanism understanding. The absorption spectra of its thin films onto quartz plates were measured before and after UV irradiation. As shown in Supplementary Fig. 42B, the absorption spectrum significantly changed after UV irradiation. The absorption peak at 318 nm disappeared after UV irradiation, which is also indicative of the decreased electronic conjugation. To gain insight into the structural change, the films of **4** were exposed to UV light for different time and then washed by CD_2Cl_2 to do the ^1H NMR characterization. The obtained ^1H NMR spectra shown in Supplementary Fig. 43 implied that UV irradiation indeed changed the chemical structure of **4** because some new peaks were observed in the aromatic proton region of the ^1H NMR spectra of the irradiated samples. Furthermore, these new resonance peaks became more and more obvious with the increase in UV irradiation time. We have tried very hard to analyze the photoreaction products. Unfortunately, the photoreaction of **4** seems to be very complicated under such a strong UV light source and the exact structures of the photoreaction products are still difficult to be determined currently. The thin-layer chromatography (TLC) analysis indicates that the photoreactions are not efficient and the resulting product is a mixture of many different structures without a concentrated point. It is very hard for us to isolate and obtain pure products to analyze their structures.

Supplementary Fig. 42 UV-vis absorption spectra of the drop-casting thin film of (A) **P1a/2a** and (B) model compound **4** before and after UV irradiation.

Supplementary Fig. 43. (A) ^1H NMR spectra of model compound **4** under different UV irradiation time and (B) their enlarged spectra. The ^1H NMR spectra were measured in CD_2Cl_2 .

In conclusion, the photo-induced RI decrease of **P1a/2a** is possibly due to the decrease in its electronic conjugation under the irradiation of strong UV light. The corresponding results and discussion are also provided in the revised manuscript and supporting information.

(7) P18: Proof that the photobleaching is due to photo-oxidative bleaching should be preferably provided, rather than the references to other polymer structures.

Response: Thanks for the reviewer's suggestion. Generally speaking, two types of photo-responsive polymers are involved for the generation of luminescent photopatterns: one type is the photo-oxidative polymers and the other type is photocrosslinkable polymers (*Chem. Soc. Rev.*, 2014, 43, 4494-4562). The possibility of photo-crosslinking process can be excluded for this polymer system because the all photopatterned films shown in Fig. 6 can be easily washed away by their good solvents tetrahydrofuran and dichloromethane. Almost no polymer residue was observed either by naked eyes or by the images taken by a fluorescent microscope (Figure R6). The absorption spectrum of **P1a/2e** and **P7** before and after UV irradiation were then measured. The results shown in **Supplementary Fig. 53** suggested that UV irradiation could lead to a decrease in the electronic conjugation of the tetraphenylethylene (TPE)-containing polymers (**P1a/2e**) and the reduced polymer (**P7**). There are many reactive sites in these polymer structures that could be photo-oxidized to result in the decreased electronic conjugation. Therefore, these polymers are possible to undergo photooxidation reactions under the irradiation of strong UV light.

Figure R6. Image of the photopatterned film of **P1a/2e** after washing by THF. The images were taken by a fluorescent microscope under 330–380 nm UV illumination.

Supplementary Fig. 53 UV-vis absorption spectra of the thin films of (A) **P1a/2e** and (B) **P7** before and after UV irradiation for 1.5 h.

Although **P1a-b/2a-c**, **P1a/2d-e** and **P7** are all composed of the repeating unit with tetraphenyl-substituted spirocyclic structure, they show different response to UV irradiation due to their detailed structural difference. As described in the manuscript, **P1a-b/2a-c** show almost no fluorescence in the solid state but the solid powder of the TPE-containing polymers (**P1a/2d-2e**) and the reduced polymer **P7** showed obvious fluorescence with λ_{em} at 504 nm, 510 nm and 463 nm, respectively (Fig. 4 and Supplementary Fig. 34). TPE is a versatile AIE building block (*J. Mater. Chem.*, 2012, 22, 23726–23740) and model compound **7** containing the AIE-active 1,2,3,4-tetraphenyl-1,3-cyclopentadiene (TPC) moiety is also demonstrated to show aggregation-enhanced emission property (Fig. 4e). The presence of these AIE-active building blocks in **P1a/2d**, **P1a/2e** and **P7** enables them to show obvious solid-state fluorescence.

The opposite photoresponsive behavior between **P1a/2d-e** and **P1a/2a** suggested that the TPE moiety may play a leading role in the photobleaching behavior of **P1a/2d-e**. It has been reported that the C=C group of TPE is photosensitive and can be photo-oxidized under intense UV irradiation (*Org. Lett.*, 2005, 7, 4265-4268) and various TPE-containing polymers have been reported to show photo-oxidative bleaching (*Macromolecules*, 2006, 39, 319-323; *Polym. Chem.*, 2013, 4, 95-105). We

also tried to oxidize the TPE film using ozone as the oxidant and study the oxidation-induced structure change in its luminescent properties. As shown in Figure R7, the strong fluorescence of TPE film was almost completely quenched after exposure to ozone (generated from an ozonator) for about 1 h. The ozonolysis reaction of the dichloromethane solution of TPE was then carried out according to the reported experimental procedures (*Eur. J. Org. Chem.* 2007, 1101-1112). After purification and structure characterization, the final oxidized product was proved to be the non-emissive benzophenone. By contrast, the non-emissive film of the tetraphenyl-substituted spirocyclic compound (**4**) was found to become fluorescent after UV irradiation (Supplementary Fig. 47A) or after the ozone treatment (Supplementary Fig. 52A). The oxidation reaction of **4** seems to produce a mixture of several different products as indicated by the TLC plate. To make a brief summary, the above experimental results indicated that the photobleaching behavior of the TPE-containing polymers may mainly result from the photooxidation of the TPE moiety.

Figure R7. (A) Photos of the thin films of TPE and (B) the associated PL spectra before and after O₃ treatment. The photos were taken under normal room light (upper panel) and 365 nm UV irradiation (lower panel), respectively.

Supplementary Fig. 47 PL spectra of the drop-casting thin film of (A) model compound **4** and (B) model compound **7** before and after UV irradiation.

Supplementary Fig. 52 (A) Photos of the thin films of model compound **4** before and after O₃ treatment taken under normal room light (upper panel) and 365 nm UV irradiation (lower panel), respectively. (B) UV spectra of model compound **4** before and after O₃ treatment. (C) ¹H NMR spectra of model compound **4** before and after O₃ treatment for 1h. The ¹H NMR spectra were measured in CDCl₃.

To investigate the possible reason for the photobleaching of **P7**, the photoresponsiveness of model compound **7** was examined. After the intense UV irradiation, model compound **7** also showed a phenomenon of UV-quenched fluorescence (Supplementary Fig. 47B). The results shown in Supplementary Fig. 51A-B and D suggested that compound **7** can also undergo photochemical reactions that lead to a decrease in its electronic conjugation under the intense UV irradiation. Some new resonance peaks were detected in the aromatic proton region of the ¹H NMR spectrum of the irradiated sample (Supplementary Fig. 51B). Because 1,3-cyclopentadiene has been widely reported to be a photosensitive unit that can undergo photooxidation reactions to afford oxidized products with less conjugated structures (*Tetrahedron*, 2018, 74, 3107-3112; *Phys. Chem. Chem. Phys.*, 2000, 2, 109-114; *Int. J. Photoenergy*, 1999, 1, 41-47), it is reasonable to propose that the photobleaching behavior of **P7** may mainly result from the photooxidation of its AIE-active 1,2,3,4-tetraphenyl-1,3-cyclopentadiene moiety.

Supplementary Fig. 51 (A and B) ^1H NMR spectra of **7** before and after UV irradiation. Detailed procedures: the film of **7** was exposed to UV light for 1.5 h and then washed by CDCl_3 to measure the ^1H NMR spectrum. (C) ^1H NMR spectrum of **4** after UV irradiation. All the ^1H NMR spectra were measured in CDCl_3 . (D) Absorption spectra of model compound **7** before and after UV irradiation for 1.5 h.

Based on the above analysis, the photobleaching phenomenon of **P1a/2d-e** and **P7** could be attributed to the occurrence of photo-oxidative bleaching process of their AIE moieties. The corresponding results and discussion are also provided in the revised manuscript and supporting information.

(8) P18: Why do **P1a/2a** and **P1b/2a** show UV-activated fluorescence? This has not been addressed and an explanation should be provided.

Response: Thanks for the reviewer's question and suggestion. The photoirradiation process was conducted in air using a strong UV light source (the incident light intensity was $\sim 18.5 \text{ mW cm}^{-2}$ and the applied power of the Mercury Arc Lamp was 180 W). There are multiple photoreactive sites in the structures of **P1a/2a** and **P1b/2a** such as the carbonyl group, carbonyl-activated $\text{C}=\text{C}$ group and the $\text{C}=\text{C}$ groups in the tetraphenyl-1,3-cyclopentadiene units. Therefore, they are possible to undergo photochemical reactions under the irradiation of such a strong UV light source. The UV-activated fluorescence of **P1a/2a** and **P1b/2a** may be due to the occurrence of photochemical reactions that can generate new structures with obvious fluorescence in the solid state. To investigate the working mechanism of the present polymer system, we first washed the photopatterned films of **P1a/2a** and **P1b/2a** shown in Fig. 6 by their good solvents such as tetrahydrofuran and dichloromethane. All these photopatterned films can be easily washed away without any residue, indicating that no cross-linking reaction occurred under UV irradiation.

Considering the intrinsic complexity and polydispersity of polymers, we then investigated the UV responsiveness of model compound **4** and **7** to assist the mechanistic study. As shown in **Supplementary Fig. 47**, the PL intensity of **4** was

obviously enhanced after UV irradiation whereas the fluorescence of **7** was quenched by UV light, which is consistent with their corresponding polymers. The opposite photoresponsive behavior between **P1a/2a** (**4**) and **P7** (**7**) also suggested that the carbonyl-activated C=C double bonds of **P1a/2a** and **P1b/2a** may play a crucial role in their UV-activated fluorescence.

Supplementary Fig. 47 PL spectra of the drop-casting thin film of (A) model compound **4** and (B) model compound **7** before and after UV irradiation.

The absorption spectra of **P1a/2a** and **4** films before and after UV irradiation (Supplementary Fig. 42) suggested that the intense UV irradiation may result in the decreased electronic conjugation of **P1a/2a** and **4**. To gain insight into the structural change, the films of **4** were exposed to UV light and then washed to measure the ^1H NMR spectra. As shown in Supplementary Fig. 43, some new peaks were observed in the aromatic proton region of the ^1H NMR spectra of the irradiated samples, which implied that UV irradiation indeed changed the chemical structure of **4**.

Supplementary Fig. 42 UV-vis absorption spectra of the drop-casting thin film of (A) **P1a/2a** and (B) model compound **4** before and after UV irradiation.

Supplementary Fig. 43. (A) ^1H NMR spectra of model compound **4** under different UV irradiation time and (B) their enlarged spectra. The ^1H NMR spectra were measured in CD_2Cl_2 .

We have tried very hard to analyze the photoreaction products. Unfortunately, the photoreaction of **4** seems to be very complicated under the strong UV irradiation and the exact structures of the photoreaction products are still difficult to be determined currently. The TLC analysis of the UV-irradiated sample showed a series of luminescent points under 365-nm UV illumination, indicating that the resulting product is a mixture of many different structures. Moreover, these points were very dilute and no new concentrated point was observed in the TLC plate, which suggested that the photoreactions are not very efficient. Therefore, it is very hard for us to isolate and obtain pure products to analyze their structures.

Ketone structures are possible to undergo photochemical reactions with the generation of free radical intermediates (*J. Am. Chem. Soc.* **1976**, *98*, 1218; *Tetrahedron Lett.* **1982**, *23*, 4531; *J. Am. Soc. Mass. Spectr.* **2011**, *22*, 2021; *ChemPhysChem* **2002**, *3*, 57). To test whether free radical species was involved in the photochemical reaction of this carbonyl-containing spirocyclic system, a radical marker 2,2,6,6-tetramethylpiperidin-1-yl)oxyl (TEMPO) was used in the solution of **4** for UV irradiation. The ^1H NMR spectrum of the sample with the addition of TEMPO did not show the above-mentioned new peaks after UV irradiation (Supplementary Fig. 48). This result indicated that the photochemical reaction of this carbonyl-containing spirocyclic system may involve free radical intermediates. The generated free radicals can be immediately trapped by TEMPO and thus prevent the occurrence of further reaction.

Supplementary Fig. 48 ^1H NMR spectra of the model compound **4** (A) after irradiating its toluene solution by UV light for 3 h, (B) after irradiating its TEMPO-containing toluene solution by UV light for 3 h, and (C) without UV irradiation. The ^1H NMR spectra were measured in CDCl_3 .

To test whether the carbonyl-activated C=C group in **4** and polymers can undergo photodimerization like coumarin derivatives (*J. Org. Chem.*, 2003, 68, 7386), we tried to stir the toluene solution and aqueous suspension of model compound **4** under the same UV irradiation conditions for about 3.5 h. Then the solvent was removed and the dried product was dissolved in CDCl_3 to do the ^1H NMR characterization. As depicted in Figure R8, the ^1H NMR results are similar to those shown in Supplementary Fig. 43. Some new resonance peaks appeared in the aromatic proton region after UV irradiation but no signals related to the resonances of dimers was observed. These results indicated that the photochemical reaction of the present spirocyclic system may not be photodimerization.

Figure R8. (A and B) ^1H NMR spectra of the solution of model compound **4** after UV irradiation for about 3.5 h. Detailed procedures: the toluene solution and the aqueous suspension of model compound **4** (prepared by adding 0.5 mL THF solution of **4** into 3 mL water under stirring) in quartz cells were irradiated by UV light for 4 h. Afterward, the solvent was removed and the solid was dissolved to measure the ^1H NMR. The ^1H NMR spectra were measured in CDCl_3 .

The photoreaction products were also analyzed by the attenuated total reflectance Fourier transform infrared (ATR-FTIR) and X-ray photoelectron spectroscopy (XPS) techniques. As shown in Figure R9, the ATR-FTIR spectra of the film of model compound **4** before and after 40-min UV irradiation seem to be similar and provide little information on the structure change. However, the high-resolution XPS spectra of **P1a/2a** film clearly revealed the change in the surface of the sample after UV irradiation. As depicted in **Supplementary Fig. 49**, the C 1s bands of the samples are deconvoluted into three peaks at 285.0 eV, 286.6 eV and 289.0 eV corresponding to C-C/C=C, C-O and C=O, respectively. By comparison, the content of the C-O and C=O signal became higher after UV irradiation and meanwhile the proportion of the C-C/C=C decreased. This result suggested that photooxidation reaction occurred after irradiating the **P1a/2a** film under strong UV light. The O 1s band of **P1a/2a** contains two peaks at 532.1 and 533.3 eV for C=O and C-O, respectively. The relative content of C=O and C-O in the C 1s and O 1s spectra indicated that the photooxidation process may generate both the C=O and C-O groups in the product structures.

Figure R9. IR spectra of the film of model compound **4** on a silicon wafer before and after 40-min UV irradiation measured using an ATR-FTIR spectrometer.

Supplementary Fig. 49 High-resolution XPS spectra of C 1s and O 1s of **P1a/2a** film before and after UV irradiation. (B and E) XPS spectra before UV irradiation. (C and F) XPS spectra after 40-min UV irradiation.

Encouraged by this result, we then tried to investigate the effect of oxygen on the photochemical reactions by conducting the photoirradiation process of the film of **4** under air and under N₂, respectively. The films of **4** were exposed to UV light for 1.5 h and then washed by CDCl₃ to do the ¹H NMR characterization and high-resolution mass spectroscopy (HRMS). The ¹H NMR results shown in **Supplementary Fig. 50** suggested that the presence of oxygen is favorable for the proceeding of photoreactions. The resonance signal of the carbonyl-activated alkene proton of **4** (at $\sim\delta$ 6.16) was obviously weakened after UV irradiation under air, and meanwhile some new resonance peaks were detected in the aromatic proton region. These results indicated that the polymers are possible to undergo photooxidation reactions with the participation of the carbonyl-activated C=C double bonds under the intense UV irradiation. The HRMS (MALDI-TOF) results shown in Figure R10 and R11 also indicates the formation of new structures after UV irradiation although the exact structures are difficult to be identified.

Supplementary Fig. 50 ¹H NMR spectra of model compound **4** before UV irradiation and after UV irradiation under air and under N₂, respectively. The ¹H NMR spectra were measured in CDCl₃.

Figure R10. HRMS (MALDI-TOF) spectrum of model compound **4** after UV irradiation under N₂ for 1.5 h.

Figure R11. HRMS (MALDI-TOF) spectrum of model compound **4** after UV irradiation under air for 1.5 h.

To investigate the role of other structural moieties except the carbonyl-activated C=C double bond on the photoresponsive properties of the polymers, the photoresponse of model compound **7** was also examined. The results shown in Supplementary Fig. 51 suggested that compound **7** can also undergo photochemical reactions under the intense UV irradiation. Some new resonance peaks were detected in the aromatic proton region of the ¹H NMR spectrum of the irradiated sample although the new signals seem to be less than those of **4**. These results suggested that the photochemical reaction of this unique multisubstituted spirocyclic structure may involve multiple reactive sites and should be a complicated process.

Supplementary Fig. 51 (A and B) ¹H NMR spectra of **7** before and after UV irradiation. Detailed procedures: the film of **7** was exposed to UV light for 1.5 h and then washed by CDCl₃ to measure the ¹H NMR spectrum. (C) ¹H NMR spectrum of **4** after UV irradiation. All the ¹H NMR spectra were measured in CDCl₃. (D) Absorption spectra of model compound **7** before and after UV irradiation for 1.5 h.

To further testify the possible photooxidation mechanism, we also tried to oxidize the film of **4** using ozone as the oxidant and investigate the change in its photophysical properties. The thin film was fabricated by spin-coating the 1,2-dichloroethane solution of **4** onto the silicon wafers. As shown in **Supplementary Fig. 52A**, the non-emissive film of **4** became fluorescent after the film was exposed to ozone (generated from an ozonator) for about 1 h. The changes in the UV and ¹H NMR spectrum of **4** after O₃ treatment (**Supplementary Fig. 52B and C**) were similar to the results obtained after UV irradiation (**Supplementary Fig. 42, 43 and 50**). These results implied that the UV-activated fluorescence of **P1a/2a** and **P1b/2a** may be due to the generation of luminescent species from their photooxidation reactions.

Supplementary Fig. 52 (A) Photos of the thin films of model compound **4** before and after O₃ treatment taken under normal room light (upper panel) and 365 nm UV irradiation (lower panel), respectively. (B) Absorption spectra of model compound **4** before and after O₃ treatment. (C) ¹H NMR spectra of model compound **4** before and after O₃ treatment for 1 h. The ¹H NMR spectra were measured in CDCl₃.

A UV/O₃ cleaner was employed to further study the synergistic effect of UV irradiation and oxidant on the photoreactions of this complex spirocyclic system. The film of model compound **4** was treated by the UV/O₃ cleaner for 1 h. The results shown in Figure R12 suggested a more obvious structure change of model **4** after UV/O₃ treatment. The peaks associated with the resonances of the protons in “a” and “b” positions of **4** significantly decreased after treatment and meanwhile the new peaks in the aromatic region become stronger, suggesting that the UV irradiation could promote the oxidation process. Moreover, the obviously weakened peak of “a” proton of **4** after UV treatment indicated that the carbonyl-activated C=C should be changed to other structures (Figure R12B). As described in the manuscript, the poor fluorescence of **P1a/2a** and **P1b/2a** in the solid state could result from the photoinduced electron transfer (PET) process (Fig. 4 and Supplementary Fig. 34). Therefore, once the PET process was blocked by the UV irradiation, the corresponding products are possible to emit strong fluorescence.

Figure R12. (A) Absorption spectra and (B) ^1H NMR spectra of model compound **4** before and after treatment using a UV/O₃ cleaner for 1 h. The ^1H NMR spectra were measured in CDCl₃.

In summary, the UV-activated fluorescence of **P1a/2a** and **P1b/2a** may be due to the occurrence of complicated photooxidation reactions with the involvement of radical species to generate new structures with obvious fluorescence in the solid state. The photochemical reactions of this complex spirocyclic structure should involve the carbonyl-activated C=C double bond together with some other reactive sites. Among them, the carbonyl-activated C=C double bonds may play a crucial role in the UV-activated fluorescence of **P1a/2a** and **P1b/2a**. As discussed in the original manuscript, the poor solid-state fluorescence of **4** and **P1a-b/2a-c** may result from the PET process. Therefore, the UV-activated fluorescence of **P1a/2a** and **P1b/2a** might be due to the block of the PET process by the photoreactions. The corresponding results and discussions were also provided in the revised manuscript and supporting information.

(9) P22: Replace ‘Discussion’ with ‘Conclusions’

Response: Thanks for the reviewer’s suggestion. We have corrected it accordingly in the revised manuscript.

Reply to the Comments of Reviewer 2

Comments: Tang and co-workers report on a new step-growth polymerization for the synthesis of photomodulatable polymers. The work appears to have been carefully performed however the novelty and impact of the work does not seem best suited for Nature Communications. This work seems better suited for a polymer specific journal.

=====

Response: We greatly appreciate the reviewer's comments on our work. As claimed in the manuscript, the development of facile and efficient synthetic strategies toward functional polymers with unique structures, attractive properties and advanced applications is of both academic interest and practical implication. This work reported a straightforward polyspiroannulation method that can transform the readily available and inexpensive 2-naphthols and diynes into valuable photoresponsive spiro-polymers with complex structures and multifunctionalities. These spiro-polymers are hard to be prepared by other polymerization methods. Unlike traditional homogeneous two-component polymerizations, one of the monomers in this stepwise polyspiroannulation is monofunctional and the presence of excess monofunctional monomers was found to be favorable for improving the polymerization efficiency. The investigation on this polyspiroannulation is expected to be instructive for the further development of other C–H-activated stepwise polymerizations of monofunctional monomers with diynes to benefit the facile preparation of new polymeric materials with diverse structures and functionalities.

In addition to the new polymerization method, the polymers described in this work possess unique multisubstituted spirocyclic structures to endow them with various interesting properties and different advanced applications. For example, these polymers show high thermal and morphological stability and good solution processibility due to the presence of rigid spirocyclic segments with twisted conformations. Some of the polymers can show efficient solid-state emission due to the effect of aggregation-enhanced emission. These excellent properties enable the polymers to find potential applications as light-emitting coating materials in photoelectric devices. The high and UV-tunable refractive index and small chromic dispersion of these spirocyclic polymers make them promising coating materials in advanced optical display systems such as optical data storage devices, gradient-index optics and integrated photonics technology. The photoresponsive fluorescent properties of the polymer thin films make them useful in data-encoding-reading, anti-counterfeiting, and organic display devices, etc. For instance, well-resolved and stable fluorescent photopatterns with both “turn-off” and “turn-on” modes can be readily fabricated based on the photobleaching and photoactivation process of different polymer films. Moreover, taking advantage of their photoresponsive refractive index, we successfully applied the polymer thin films in integrated silicon photonics techniques and achieved the permanent modification of resonance wavelengths of microring resonators by UV irradiation. This proof-of-concept study which will benefit the progress of both the photoresponsive polymers and the integrated silicon photonics technique. The applicability could be further strengthened

by developing polymer materials with a larger refractive index change in response to UV irradiation or using polymer substrates to extend the application to flexible photonics. In short, this work not only is helpful for polymer chemists but also provide useful information for material scientists to develop novel functional materials with attractive photoluminescent or optical properties.

We do hope the straightforward synthetic strategy, careful study of the structure-property relationship and fascinating functionalities of the obtained polymers in this work can promote further research on the design, construction, modification, and functionalization of diverse polymeric materials. *Nature Communications* is a multidisciplinary journal that publishes high-quality research from all areas of the natural sciences and aim to represent important advances of significance to specialists within each field. Therefore, we believe this work should be of interest to the readers of *Nature Communications*.

(1) The authors claim term a monomer nonstoichiometric strategy, but this term is somewhat misleading. Although the authors use a nonstoichiometric quantity of monomers, this optimization is mostly to match the catalyst reactivity for the synthesis of higher molecular weight polymers. This reviewer suggests using different terminology to describe their approach.

Response: Thanks for the reviewer's comment and suggestion. We are sorry about the unintentional misleading. To distinguish the effective stoichiometry from the feed ratio of monomers, "apparent nonstoichiometry in monomer ratio" or "without the constraint of apparent stoichiometric balance in monomers" were used to describe this approach in the revised manuscript. The changes we made are highlighted in red for easy identification.

(2) The authors then characterize the polymers and their optical properties. Although many interesting results are reported, the photomodulatable properties are reported as a broad collection of highlights of results that this reviewer believes would be better presented if they were presented more thoroughly in a series of separate manuscripts.

Response: Thanks for the reviewer's comment and suggestion. The photomodulatable properties including the UV-responsive fluorescence and refractive index as well as their associated applications are closely related to the unique multisubstituted spiro-polymer structures generated by this C-H activated polyspiroannulation method. We believe it would be better to present the synthesis, structures, properties, and potential applications of the polymers together in one comprehensive paper to provide readers with a relatively complete picture. Some detailed discussions about the investigation on the structure-property relationship of the polymers were also provided in the revised manuscript and supporting information to make this paper more thoroughly presented. The changes we made are highlighted in red for easy identification.

Reply to the Comments of Reviewer 3

Comments: The authors reported the synthesis of a series of photomodulatable spirocyclic polymers using nonstoichiometric stepwise polymerization and investigated their optical and thermal properties. They also use these polymers to fabricate fluorescent photopatterns and make integrated silicon photonics. A lot of data and interesting results were provided, but the corresponding discussions were not enough. This paper might be suitable for publication in this high-level journal after the following issues were settled.

Response: We greatly appreciate the positive comments from the reviewer on our work. Accordingly, some more in-depth discussions were added in the revised paper. Our point-by-point response is given below.

(1) In Figure 4f, the curve of compound **7** showed a turning point and following sharp increase, while the other polymers only showed a slow variation trending. Please explore the possible reason.

Response: We appreciate the reviewer's careful reading of our manuscript to raise this good question. The fluorescence of model compound **7** was not obviously changed until a large amount of water ($f_w = 80\%$ and 90%) was added, showing a turning point followed by a sharp increase. On the other hand, the photoluminescence (PL) of polymers is more sensitive to the change in the solvent environment and shows a slow variation trending. The faint emission of the polymer solutions can be enhanced even at low water fractions. The higher the water content, the stronger is the PL intensity. This phenomenon may originate from the more hydrophobic nature of polymers, which makes them easier to form aggregates in aqueous mixtures with low water fractions (*Polym. Chem.*, 2016, 7, 2501). To verify our hypothesis, we conducted the particle size analysis for compound **7** and the polymers using dynamic light scattering. Taking **P1a/2e** as an example, the results demonstrated the formation of nanoparticles with an average diameter of 185, 322 and 306 nm in its aqueous mixtures with a low water fraction of 10, 30 and 50%, respectively. Also, nanoparticles with an average diameter of 424, 537 and 377 nm were detected in the aqueous mixtures of **P1a/2e** with a higher water fraction of 70, 80 and 90%, respectively. By contrast, model compound **7** can form large aggregates only at high water fractions. Nanoparticles with an average diameter of 362 and 424 nm were detected in its 80 and 90% aqueous mixtures, respectively.

From the perspective of AIE mechanism, the AIE effect of **7**, **P7**, **P1a/2d**, and **P1a/2e** is proposed to be associated with the restriction of intramolecular motion (RIM) in the aggregated state (*Adv. Mater.*, 2014, 26, 5429; *Chem. Rev.*, 2015, 115, 11718). The phenyl rings in **7** are linked to carbon-carbon double bonds and can undergo rotation freely in the solution state. Such active intramolecular rotation will

act as a non-radiative pathway for the excitons to decay to the ground state, rendering **7** poorly emissive in pure THF solution and aqueous mixtures with a low water fractions. Upon aggregate formation, such motion is restricted due to the physical constraint. This blocks the non-radiative relaxation pathway and thus enhances the emission of the luminogen. Since the luminogen units in polymers are linked together by covalent bonds, their rotation has been partially restricted in nature. The rigid structure of polymers could also attribute to the formation of aggregates in aqueous mixtures with low water fractions. The RIM process of polymers can be activated with the formation of aggregates at low water fractions. Therefore, the AIE curves of polymers showed slow variation trending, while compound **7** showed a turning point and following sharp increase. The corresponding discussions were provided in the revised manuscript.

(2) On line 346 (page 16), they ascribed the change in refractive properties to the photosensitive C=C and carbonyl groups. Please indicate the possible photo-reaction of these polymers under UV irradiation.

Response: Thanks for the reviewer's suggestion. There are multiple photoreactive sites in the polymer structures such as the carbonyl group, carbonyl-activated C=C group and the C=C groups in the tetraphenyl-1,3-cyclopentadiene units. Under the irradiation of the strong UV light source (the incident light intensity was $\sim 18.5 \text{ mW cm}^{-2}$ and the applied power of the Mercury Arc Lamp was 180 W), the present polymers are possible to undergo some photochemical reactions. Indeed, the UV-vis absorption and ^1H NMR spectra of **P1a/2a** and model compound **4** shown in **Supplementary Fig. 42 and 43** demonstrated the occurrence of photochemical reactions and the change in the chemical structures under UV irradiation. The absorption results also indicated the decrease in the electronic conjugation of **P1a/2a** and model compound **4** after UV irradiation.

Supplementary Fig. 42 UV-vis absorption spectra of the drop-casting thin film of (A) **P1a/2a** and (B) model compound **4** before and after UV irradiation.

Supplementary Fig. 43. (A) ^1H NMR spectra of model compound **4** under different UV irradiation time and (B) their enlarged spectra. The ^1H NMR spectra were measured in CD_2Cl_2 .

We have tried very hard to analyze the photoreaction products. Unfortunately, the photoreaction seems to be very complicated under such a strong UV light source and the exact structures of the photoreaction products are still difficult to be determined currently. The thin-layer chromatography (TLC) analysis of the UV-irradiated sample of **4** indicates that the photoreactions are not efficient and the resulting product is a mixture of many different structures without a concentrated point. It is very hard for us to isolate and obtain pure products to analyze their structures. The relevant experiments that we have tried and the corresponding results are provided as follows.

To investigate the possible photo-reactions of these polymers, the UV-irradiated polymer films were first washed by their original good solvents such as tetrahydrofuran and dichloromethane. All these UV-irradiated polymer films can be washed without any residue, indicating that no cross-linking reaction occurred under UV irradiation.

Because both the carbonyl group and the $\text{C}=\text{C}$ group are potential to be photosensitive, we then tried to study the effect of carbonyl group and the $\text{C}=\text{C}$ group on the refractive properties of the polymers, respectively. The reduction reaction of **4** and **P1a/2a** was carried out in the presence of lithium aluminum hydride (LiAlH_4). As shown in Fig. 4a and 4b, compound **7** and polymer **P7** without the carbonyl-activated $\text{C}=\text{C}$ group were successfully obtained in high conversion rates. We then measured the RI of **P7** and compared it with that of **P1a/2a** (Supplementary Table 6 and Supplementary Fig. 41). The results showed that the RI of **P7** is lower than that of **P1a/2a**, which indicated that the conversion of the unsaturated $\text{C}=\text{C}$ to saturated $\text{C}-\text{C}$ can lead to a decrease in RI of the corresponding polymer due to the decreased electronic conjugation.

Supplementary Fig. 41 Wavelength dependence of refractive indices of thin films of P1a/2a and P7.

Unfortunately, the carbonyl group in this structure is hard to be reacted to assist the mechanistic understanding. For example, we have tried the ketalation of model compound **4** and **7** but failed to get the products. We have also tried to transform the C=O of model compound **4** via reduction reaction using sodium borohydride (NaBH_4), $\text{NaBH}_4/\text{CeCl}_3 \cdot 7\text{H}_2\text{O}$ (Luche reduction), and LiAlH_4 as the reductant, respectively, but these reductants mainly react with the carbonyl-activated C=C rather than the C=O group. Furthermore, the reduction reaction of model compound **7** also cannot take place with LiAlH_4 as the reductant. The low reactivity of the carbonyl group is possibly due to its large steric hindrance in this multi-substituted spirocyclic system.

As ketone structures are possible to undergo photochemical reactions with the generation of free radical intermediates (*J. Am. Chem. Soc.* **1976**, *98*, 1218; *Tetrahedron Lett.* **1982**, *23*, 4531; *J. Am. Soc. Mass. Spectr.* **2011**, *22*, 2021; *ChemPhysChem* **2002**, *3*, 57), we employed a radical marker 2,2,6,6-tetramethylpiperidin-1-yl)oxyl (TEMPO) to test whether the photochemical reaction of the present system involves free radical intermediates. The ^1H NMR spectrum of the sample with the addition of TEMPO did not show new peaks in the aromatic proton region after UV irradiation (Supplementary Fig. 48). This result indicated that the photochemical reaction of this carbonyl-containing spirocyclic system may involve free radical intermediates. The generated free radicals can be immediately trapped by TEMPO and thus prevent the occurrence of further reaction.

Supplementary Fig. 48 ^1H NMR spectra of the model compound **4** (A) after irradiating its toluene solution by UV light for 3 h, (B) after irradiating its TEMPO-containing toluene solution by UV light for 3 h, and (C) without UV irradiation. The ^1H NMR spectra were measured in CDCl_3 .

We also tried to analyze the photoreaction products by X-ray photoelectron spectroscopy (XPS). As shown in **Supplementary Fig. 49**, the high-resolution XPS spectra of **P1a/2a** film clearly revealed the change in the surface of the sample after UV irradiation. The C 1s bands of the samples are deconvoluted into three peaks at 285.0 eV, 286.6 eV and 289.0 eV corresponding to C-C/C=C, C-O and C=O, respectively. By comparison, the content of the C-O and C=O band became higher after UV irradiation and meanwhile the proportion of the C-C/C=C decreased. This result suggested that photooxidation reaction occurred after irradiating the **P1a/2a** film under strong UV light. The O 1s band of **P1a/2a** contains two peaks at 532.1 and 533.3 eV for C=O and C-O, respectively. The relative content of C=O and C-O band in the C 1s and O 1s spectra indicated that the photooxidation process may generate both the C=O and C-O groups in the product structures.

Supplementary Fig. 49 High-resolution XPS spectra of C 1s and O 1s of P1a/2a film before and after UV irradiation. (B and E) XPS spectra before UV irradiation. (C and F) XPS spectra after 40-min UV irradiation.

Encouraged by this result, we then tried to investigate the effect of oxygen on the photochemical reactions by conducting the photoirradiation process of the film of **4** under air and under N₂, respectively. The films of **4** were exposed to UV light for 1.5 h and then washed by CDCl₃ to do the ¹H NMR characterization and high-resolution mass spectroscopy (HRMS). The ¹H NMR results shown in Supplementary Fig. 50 suggested that the presence of oxygen is favorable for the proceeding of photoreactions. The resonance signal of the carbonyl-activated alkene proton of **4** (at $\sim\delta$ 6.16) was obviously weakened after UV irradiation under air, and meanwhile some new resonance peaks were detected in the aromatic proton region. These results indicated that the polymers are possible to undergo photooxidation reactions with the participation of the carbonyl-activated C=C double bonds under the intense UV irradiation.

Supplementary Fig. 50 ^1H NMR spectra of model compound **4** before UV irradiation and after UV irradiation under air and under N_2 , respectively. The ^1H NMR spectra were measured in CDCl_3 .

To test whether the carbonyl-activated $\text{C}=\text{C}$ group in **4** and polymers can undergo photodimerization like coumarin derivatives (*J. Org. Chem.*, 2003, 68, 7386), we tried to stir the toluene solution and aqueous suspension of model compound **4** under the same UV irradiation conditions for about 3.5 h. Then the solvent was removed and the dried product was dissolved in CDCl_3 to do the ^1H NMR characterization. As depicted in Figure R8, the ^1H NMR results are similar to those shown in Supplementary Fig. 43. Some new resonance peaks appeared in the aromatic proton region after UV irradiation but no signals related to the resonances of dimers was observed. These results indicated that the photochemical reaction of the present spirocyclic system may not be photodimerization. In addition, we have also tried to irradiate compound **7** (without the carbonyl-activated $\text{C}=\text{C}$ bond) to examine the role of other groups. The results shown in Supplementary Fig. 51 suggested that compound **7** can also undergo photochemical reactions and lead to the decreased electronic conjugation. Some new resonance peaks were also detected in the aromatic proton region of the ^1H NMR spectrum of **7** after UV irradiation although the new signals seem to be less than those of **4**. Therefore, the photochemical reaction of this multi-substituted spirocyclic structure may involve multiple reactive sites and should be a complicated process.

Figure R8. (A and B) ^1H NMR spectra of the solution of model compound **4** after UV irradiation for about 3.5 h. Detailed procedures: the toluene solution and the aqueous suspension of model compound **4** (prepared by adding 0.5 mL THF solution of **4** into 3 mL water under stirring) in quartz cells were irradiated by UV light for 4 h. Afterward, the solvent was removed and the solid was dissolved to measure the ^1H NMR. The ^1H NMR spectra were measured in CDCl_3 .

Supplementary Fig. 51 (A and B) ^1H NMR spectra of **7** before and after UV irradiation. Detailed procedures: the film of **7** was exposed to UV light for 1.5 h and then washed by CDCl_3 to measure the ^1H NMR spectrum. (C) ^1H NMR spectrum of **4** after UV irradiation for 3 h. All the ^1H NMR spectra were measured in CDCl_3 . (D) Absorption spectra of model compound **7** before and after UV irradiation for 1.5 h.

In conclusion, the present polymers are possible to undergo complicated photooxidation reactions with the involvement of radical species under the irradiation of strong UV light. These photoreactions could result in the decrease of the electronic conjugation of the polymers and thus lead to the photo-induced RI decrease. To explain the change in refractive properties more appropriately, the manuscript was revised as follows: “The results shown in Supplementary 41–43 suggest that such a large change in refractive index is possibly due to the UV irradiation-induced structure change of the polymers. There are multiple reactive sites in the polymer structures that could be photo-oxidized under the irradiation of strong UV light, which

will reduce the electronic conjugation and thus lead to the decrease in their refractive indices.” The corresponding results and discussions were also provided in the revised supporting information.

(3) In Figure 6c, P1a/2a and P1b/2a show an interesting phenomenon of UV-activated fluorescence. Please explain this unusual phenomenon and the possible UV-induced structural change.

Response: Thanks for the reviewer’s question. The UV-activated fluorescence of P1a/2a and P1b/2a may be due to the occurrence of photochemical reactions that can generate new structures with obvious fluorescence in the solid state. As shown in Fig. 4 and Supplementary Fig. 34, P1a/2a and P1b/2a are almost non-fluorescent in the solid state, which may result from the photoinduced electron transfer (PET) process. Once the PET process is blocked, the corresponding products (7 and P7) can emit strong fluorescence under UV illumination. These results indicated that the block of the PET process of P1a/2a and P1b/2a by the photoreactions is also possible to turn-on their fluorescence. Due to the intrinsic complexity and polydispersity of polymers, the UV responsiveness of model compound 4 and 7 was investigated to assist the mechanistic study. As shown in Supplementary Fig. 47, the PL intensity of 4 was obviously enhanced after UV irradiation whereas the fluorescence of 7 was quenched by UV light, which is consistent with their corresponding polymers. The opposite photoresponsive behavior between P1a/2a (4) and P7 (7) also suggested that the carbonyl-activated C=C double bonds may play a crucial role in the UV-activated fluorescence of P1a/2a and P1b/2a.

Supplementary Fig. 47 PL spectra of the drop-casting thin film of (A) model compound 4 and (B) model compound 7 before and after UV irradiation.

The photopatterned films of P1a/2a and P1b/2a can be easily washed off without any residue by their good solvents such as tetrahydrofuran and dichloromethane, indicating that no cross-linking reaction occurred under UV irradiation. The UV-vis

absorption spectra of P1a/2a and 4 films before and after UV irradiation (Supplementary Fig. 42) suggested that the intense UV irradiation may result in an obvious decrease in the electronic conjugation of P1a/2a and 4. To gain insight into the possible UV-induced structural change, the UV-irradiated films of 4 were washed to measure their ^1H NMR spectra. As shown in Supplementary Fig. 43, some new peaks were observed in the aromatic proton region of the ^1H NMR spectra of the irradiated samples, which implied that UV irradiation indeed changed the chemical structure of 4. We have tried very hard to analyze the photoreaction products. Unfortunately, the photoreaction of 4 seems to be very complicated under the strong UV irradiation and the exact structures of the photoreaction products are still difficult to be determined currently. The TLC analysis indicates that the photoreactions are not very efficient as no new concentrated point was observed in the TLC plate, and the resulting product seems to be a mixture of many different structures. It is very hard for us to isolate and obtain pure products to analyze their structures.

Supplementary Fig. 42 UV-vis absorption spectra of the drop-casting thin film of (A) P1a/2a and (B) model compound 4 before and after UV irradiation.

Supplementary Fig. 43. (A) ^1H NMR spectra of model compound 4 under different UV irradiation time and (B) their enlarged spectra. The ^1H NMR spectra were measured in CD_2Cl_2 .

The ^1H NMR results shown in Supplementary Fig. 48 and Supplementary Fig. 51 implied that the photochemical reaction of this multi-substituted spirocyclic structure may involve multiple reactive sites and should be a complicated process with the involvement of free radical intermediates. As shown in Supplementary Fig. 48, the ^1H NMR spectrum of the **4** with the addition of a radical marker (TEMPO) did not show the above-mentioned new peaks after UV irradiation. This result indicated that the photochemical reaction of this carbonyl-containing spirocyclic system may involve free radical intermediates. The generated free radicals can be immediately trapped by TEMPO and thus prevent the occurrence of further reaction. The ^1H NMR spectrum of the irradiated compound **7** (without the carbonyl-activated C=C double bond) also showed new resonance peaks in the aromatic proton region (Supplementary Fig. 51). This result suggested that besides the carbonyl-activated C=C double bond, the other reactive sites of this unique multisubstituted spirocyclic structure may also be reacted under the irradiation of strong UV light.

Supplementary Fig. 48 ^1H NMR spectra of the model compound **4** (A) after irradiating its toluene solution by UV light for 3 h, (B) after irradiating its TEMPO-containing toluene solution by UV light for 3 h, and (C) without UV irradiation. The ^1H NMR spectra were measured in CDCl_3 .

Supplementary Fig. 51 (A and B) ¹H NMR spectra of **7** before and after UV irradiation. Detailed procedures: the film of **7** was exposed to UV light for 1.5 h and then washed by CDCl₃ to measure the ¹H NMR spectrum. (C) ¹H NMR spectrum of **4** after UV irradiation for 3 h. All the ¹H NMR spectra were measured in CDCl₃. (D) Absorption spectra of model compound **7** before and after UV irradiation for 1.5 h.

We also tried to analyze the photoreaction products by X-ray photoelectron spectroscopy (XPS). The high-resolution XPS spectra of **P1a/2a** film clearly revealed the change in the surface of the sample after UV irradiation (**Supplementary Fig. 49**). The change of the C-O and C=O band in the C 1s and O 1s spectra suggested that photooxidation reaction possibly occurred after irradiating the **P1a/2a** film under strong UV light and the photooxidation process may generate both the C=O and C-O groups in the product structures. Encouraged by this result, we then tried to investigate the effect of oxygen on the photochemical reactions by conducting the photoirradiation process of the film of **4** under air and under N₂, respectively. The ¹H NMR results shown in **Supplementary Fig. 50** suggested that the presence of oxygen is favorable for the proceeding of photoreactions. The resonance signal of the carbonyl-activated alkene proton of **4** (at $\sim\delta$ 6.16) was obviously decreased together with the appearance of some new resonance peaks in the aromatic proton region after UV irradiation under air. These results indicated that the polymers are possible to undergo photooxidation reactions with the participation of the carbonyl-activated C=C double bonds under the intense UV irradiation.

Supplementary Fig. 49 High-resolution XPS spectra of C 1s and O 1s of P1a/2a film before and after UV irradiation. (B and E) XPS spectra before UV irradiation. (C and F) XPS spectra after 40-min UV irradiation.

Supplementary Fig. 50 ^1H NMR spectra of model compound **4** before UV irradiation and after UV irradiation under air and under N_2 , respectively. The ^1H NMR spectra were measured in CDCl_3 .

Furthermore, to testify the possible photooxidation mechanism, we also tried to oxidize the film of **4** using ozone as the oxidant and investigate the change in its

photophysical properties. The thin film was fabricated by spin-coating the 1,2-dichloroethane solution of **4** onto the silicon wafers. As shown in **Supplementary Fig. 52A**, the non-emissive film of **4** became fluorescent after the film was exposed to ozone (generated from an ozonator) for about 1 h. The changes in the UV and ^1H NMR spectrum of **4** after O_3 treatment (**Supplementary Fig. 52B and C**) were similar to the results obtained after UV irradiation (**Supplementary Fig. 42, 43 and 50**). These results indicated that the UV-activated fluorescence of **P1a/2a** and **P1b/2a** may be due to the generation of luminescent species from their photooxidation reactions.

Supplementary Fig. 52 (A) Photos of the thin films of model compound **4** before and after O_3 treatment taken under normal room light (upper panel) and 365 nm UV irradiation (lower panel), respectively. (B) UV spectra of model compound **4** before and after O_3 treatment. (C) ^1H NMR spectra of model compound **4** before and after O_3 treatment for 1h. The ^1H NMR spectra were measured in CDCl_3 .

A UV/ O_3 cleaner was employed to further study the synergistic effect of UV irradiation and oxidant on the photoreactions of this complex spirocyclic system. The film of model compound **4** was treated by the UV/ O_3 cleaner for 1 h. The results shown in Figure R12 suggested a more obvious structure change of model **4** after UV/ O_3 treatment. The peaks associated with the resonances of the protons in “a” and “b” positions of **4** significantly decreased after treatment and meanwhile the new peaks in the aromatic region become stronger, suggesting that the UV irradiation could promote the oxidation process. Moreover, the obviously weakened peak of “a”

proton of **4** after UV treatment also indicated that the carbonyl-activated C=C should be changed to other structures.

Figure R12. (A) Absorption spectra and (B) ¹H NMR spectra of model compound **4** before and after treatment using a UV/O₃ cleaner for 1 h. The ¹H NMR spectra were measured in CDCl₃.

In summary, the UV-activated fluorescence of **P1a/2a** and **P1b/2a** may be due to the occurrence of complicated photooxidation reactions with the involvement of radical species to generate new structures with obvious fluorescence in the solid state. The photochemical reactions of this complex spirocyclic structure should involve the carbonyl-activated C=C double bond together with some other reactive sites although the exact structures of the photoreaction products are still difficult to be identified currently. The carbonyl-activated C=C double bonds of **P1a/2a** and **P1b/2a** may play a crucial role in their UV-activated fluorescence. As discussed above, the poor solid-state fluorescence of **4** and **P1a–b/2a–c** may result from the PET process. Therefore, the UV-activated fluorescence of **P1a/2a** and **P1b/2a** might be due to the block of the PET process by the photoreactions. The corresponding results and discussions were also provided in the revised manuscript and supporting information.

(4) Please carefully check the typos in this manuscript, for example, on line 350, “the n value at 1500 nm (n1550)”.

Response: Thanks for the reviewer’s suggestion. We have double-checked the whole manuscript carefully and corrected the typos accordingly in the revised manuscript.

REVIEWERS' COMMENTS:

Reviewer #1 (Remarks to the Author):

During revision, the authors have significantly improved the scientific quality of the manuscript. All the major concerns have been satisfactorily addressed and I am happy to support acceptance of the work for publication in Nature Communications.

Reviewer #2 (Remarks to the Author):

I believe that the authors have done an adequate job at addressing the concerns the reviewers raised and is now suitable for publication.

Reviewer #3 (Remarks to the Author):

After I checked the authors revised manuscript, the papers has been improved. Now it can be considered for acceptance.

Response to Reviewers' Comments for Manuscript NCOMMS-19-15699A

Reviewer #1 (Remarks to the Author):

Comments: During revision, the authors have significantly improved the scientific quality of the manuscript. All the major concerns have been satisfactorily addressed and I am happy to support acceptance of the work for publication in Nature Communications.

=====

Response: We sincerely thank the reviewer for spending your precious time to re-review our manuscript. Thank you for the recommendation for publication of our work.

=====

Reviewer #2 (Remarks to the Author):

Comments: I believe that the authors have done an adequate job at addressing the concerns the reviewers raised and is now suitable for publication.

=====

Response: We sincerely thank the reviewer for spending your precious time to re-review our manuscript. Thank you for the recommendation for publication of our work.

=====

Reviewer #3 (Remarks to the Author):

Comments: After I checked the authors revised manuscript, the papers has been improved. Now it can be considered for acceptance.

=====

Response: We sincerely thank the reviewer for spending your precious time to re-review our manuscript. Thank you for the recommendation for publication of our work.

=====